# CryoEM and computer simulations reveal a novel kinase conformational switch in bacterial chemotaxis signaling

C Keith Cassidy[1,2†], Benjamin A Himes[3†], Frances J Alvarez[3†], Jun Ma[3], Gongpu Zhao[3], Juan R Perilla[1,2], Klaus Schulten[1,2*], Peijun Zhang[3*]

[1]Department of Physics, University of Illinois at Urbana-Champaign, Urbana, United States; [2]Beckman Institute, University of Illinois at Urbana-Champaign, Urbana, United States; [3]Department of Structural Biology, University of Pittsburgh School of Medicine, Pittsburgh, United States

**Abstract** Chemotactic responses in bacteria require large, highly ordered arrays of sensory proteins to mediate the signal transduction that ultimately controls cell motility. A mechanistic understanding of the molecular events underlying signaling, however, has been hampered by the lack of a high-resolution structural description of the extended array. Here, we report a novel reconstitution of the array, involving the receptor signaling domain, histidine kinase CheA, and adaptor protein CheW, as well as a density map of the core-signaling unit at 11.3 Å resolution, obtained by cryo-electron tomography and sub-tomogram averaging. Extracting key structural constraints from our density map, we computationally construct and refine an atomic model of the core array structure, exposing novel interfaces between the component proteins. Using all-atom molecular dynamics simulations, we further reveal a distinctive conformational change in CheA. Mutagenesis and chemical cross-linking experiments confirm the importance of the conformational dynamics of CheA for chemotactic function.

*For correspondence: kschulte@ks.uiuc.edu (KS); pez7@pitt.edu (PZ)

†These authors contributed equally to this work

Competing interests: The authors declare that no competing interests exist.

## Introduction

Bacterial chemotaxis is a ubiquitous, two-component signal transduction system that allows cells to extract information from environmental chemical gradients and place themselves within the nutrient-optimal portion of their habitat (*Wadhams and Armitage, 2004*; *Capra and Laub, 2012*; *Eisenbach, 2004*). Though the topology and complexity of the protein networks employed in bacterial chemotaxis vary by species, each uses the histidine kinase CheA (component 1) and response regulator CheY (component 2) to set up an intracellular phosphorylation cascade that regulates the motile behavior of the cell (*Szurmant and Ordal, 2004*). CheA, in particular, is a multi-domain protein, consisting of five separate and functionally distinct domains (P1-P5): P1-phosphoryl transfer domain, P2-substrate binding domain, P3-dimerization domain, P4-kinase domain and P5-regulatory domain. In addition to CheA and CheY, an expanded set of molecules assist in the mechanics of signal reception, transmission, and regulation. Specifically, bacteria utilize dedicated chemoreceptors (also known as methyl-accepting chemotaxis proteins, MCPs) to recognize ambient chemicals and transmit mechanical signals across the cell membrane to affect CheA kinase activity (*Ortega et al., 2013*; *Parkinson et al., 2015*). The adaptor protein CheW universally participates in the coupling of conformational changes within receptors to kinase regulation (*Szurmant and Ordal, 2004*; *Liu and Parkinson, 1989*). Bacteria, moreover, have evolved the ability to tune or adapt their chemotactic sensitivity to stimulus intensity, giving rise to short-term molecular memory and allowing an appropriate system response over wide ranges of chemical concentrations (*Hazelbauer and Lai, 2010*;

**eLife digest** To survive, an organism must be able to collect and interpret information about its environment and behave accordingly. Bacteria are able to do this via a process called "chemotaxis". Inside the bacteria are sensors that contain a two-dimensional network of proteins called a chemosensory array, which detect chemical changes in the environment and signal to motor proteins to allow the bacterium to move to a more favorable location. Thousands of proteins make up the chemosensory array, and so a key question is how do these proteins interact with each other to work together as a team?

Cassidy, Himes, Alvarez et al. have now used a technique called cryo-electron tomography to determine the three-dimensional structure of the chemosensory array in the bacteria species *Escherichia coli* in high detail. This revealed the structures of several key components of the array, including some protein regions that are critical for signaling during chemotaxis.

Cassidy et al. were then able to use the electron tomography data to create a model of the array that details all of its individual atoms. Supercomputer simulations of this model revealed that during chemotaxis, a key signaling protein changes shape in a way that is critical for signal processing. This shape change was confirmed to be important for chemotaxis by chemical experiments and tests on mutant *E. coli* cells.

The next steps are to further improve the structure so that more details of the array organization can be distinguished, as well as to investigate the structure of other signaling states. By assembling structural "snapshots" of these different states, in the long term Cassidy et al. aim to develop models that detail the atoms in every one of the components involved in the chemotaxis signaling pathway.

*Parkinson et al., 2015*). In the case of the model organism, *Escherichia coli,* the adaptation mechanism involves the use of two enzymes, CheR and CheB, which reversibly modify specific residues in the receptor molecules (*Hazelbauer et al., 2008*; *Hazelbauer and Lai, 2010*; *Goy et al., 1977*; *Ortega et al., 2013*).

The tunable control of chemotactic activity requires the assembly of collaborative core-signaling units, involving the chemoreceptor trimer of dimers (TOD) (*Amin and Hazelbauer, 2010*; *Li and Hazelbauer, 2011*), CheA dimer and CheW monomer (*Li and Hazelbauer, 2011*; *Falke and Piasta, 2014*). Through the formation of large, highly organized clusters known as chemosensory arrays, thousands of core-signaling units establish a network of cooperative interactions that dramatically affect signal transmission and regulation and endow the basic two-component chemotaxis infrastructure with heightened information processing and control capabilities (*Hazelbauer and Lai, 2010*; *Falke and Piasta, 2014*; *Sourjik and Armitage, 2010*; *Bray et al., 1998*; *Tu, 2013*). Important progress has been made in the characterization of localized portions of array structure using a battery of genetic, biochemical, and biophysical techniques. This progress includes the derivation of atomic structures of the individual core signaling components (*Kim and Yokota, 1999*; *Bilwes et al., 1999*; *Park et al., 2006*; *Li et al., 2007*; *Griswold et al., 2002*) and several of their sub-complexes (*Park et al., 2006*; *Li and Bayas, 2013*; *Briegel et al., 2012*) as well as the elucidation of key interactions between the core signaling components in soluble multi-protein complexes (*Bhatnagar et al., 2010*; *Vu et al., 2012*; *Wang et al., 2012*) and in reconstituted, attractant-regulated core complexes (*Li and Hazelbauer, 2011*; *Piasta et al., 2013*; *Natale et al., 2013*; *Falke and Piasta, 2014*; *Li and Hazelbauer, 2014*).

Recently, a global view of the extended structural organization of chemosensory arrays has emerged from cryo-electron tomography (cryoET) studies of native bacterial cells (*Briegel et al., 2009; 2012*; *Liu et al., 2012*; *Zhang et al., 2007*). Specifically, chemoreceptor TODs were observed to form hexagonal arrays with a 12 nm lattice spacing conserved across several, distantly related bacterial species including *E. coli* and *T. maritima* (*Briegel et al., 2009*; *2012*; *2014a*; *Liu et al., 2012*; *Zhang et al., 2007*). The conservation of this hexagonal organization has also been demonstrated in non-membrane spanning cytoplasmic chemosensory arrays (*Briegel et al., 2014a*; *2014b*). Additionally, studies using cryoET with sub-tomogram averaging, in tandem with crystallographic structures of portions of the core complex, have reported the extended structure of the

array to consist of receptor TODs packed in a two-facing-two fashion about kinase-filled and kinase-empty rings (*Briegel et al., 2012*; *2014b*; *Liu et al., 2012*). However, due to the thickness of the cells as well as cellular crowding and heterogeneity, past cellular tomography studies have been limited to discerning only the overall arrangement of the core signaling components.

The lack of a high-resolution description of the intact and extended chemosensory array structure has hindered the development of a detailed understanding of molecular events occurring within the array during signaling. To address this problem, we have taken a joint experimental-computational approach. In particular, we have developed a novel reconstitution method yielding ultra-thin mono-layer samples of core-signaling complex arrays, from which we derived a three-dimensional density map of the reconstituted core-signaling complex at 11.3 Å resolution using cryoET and sub-tomo-gram classification and averaging. Through the computational synthesis of existing X-ray crystallog-raphy data and our new cryoET data, we have constructed an atomic model of the extended chemosensory array. Our model highlights novel interaction interfaces between the receptor, CheA, and CheW and permits the use of large-scale, all-atom molecular dynamics (MD) simulations (*Perilla et al., 2015*) to further illuminate the molecular details of a key kinase-signaling event.

## Results

### Reconstitution of bacterial chemotaxis core-signaling complex arrays

To overcome the limitations imposed by cellular tomography of native chemosensory arrays (*Briegel et al., 2009*; *2012*; *Liu et al., 2012*; *Zhang et al., 2007*), we elected to establish an in vitro reconstituted system for high-resolution structural analysis of the signaling complex. Inspired by the template-directed method to assemble functional signaling complexes on lipid vesicles (*Montefusco et al., 2007*; *Shrout et al., 2003*), we designed a $Ni^{2+}$-NTA lipid containing monolayer system (*Taylor et al., 2007*; *Taylor and Taylor, 1999*) to reconstitute the two-dimensional (2D) arrays of signaling complexes for structural analysis. To this end, we expressed and purified to high homogeneity *E. coli* chemotaxis proteins: CheA, CheW, and a His-tagged cytoplasmic signaling domain of the wild-type (wt) Tar receptor (TarCF). His-tagged TarCF can be readily incorporated into the $Ni^{2+}$-NTA lipid monolayers, seen as homogeneous particles in the EM micrographs of nega-tively stained specimen (*Figure 1A*). Only in the presence of all three components (TarCF, CheA, and CheW) were ordered arrays evident (*Figure 1B*), and even then, these microcrystalline 2D arrays were only formed under strictly constrained input ratios of the three components, a finding that is consistent with previous results indicating that the chemotactic function of the complex is diminished when one of the components is reduced or over-produced (*Zhang et al., 2007*). The optimal condi-tion for array formation was established to be a mixture of TarCF, CheA and CheW with a molar ratio of 9:18:18 µM for TarCF:CheA:CheW in a lipid monolayer containing 2:1 DOPC:DOGS-NTA-$Ni^{2+}$ lipids (33% $Ni^{2+}$-NTA lipid). Notably, the input molar ratio of the reconstitution mixture does not reflect the actual ratio of components incorporated into the monolayer, as illustrated in *Figure 1C*. The resulting arrays are organized in hexagonal lattices with 12 nm spacing (*Figure 1B* inset, white arrow), resembling the arrays formed in native cells (*Briegel et al., 2012*; *Liu et al., 2012*).

### CryoET of the chemotaxis core-signaling complex arrays

Compared to previous cellular tomography studies (*Briegel et al., 2012*; *2014b*; *Liu et al., 2012*; *Zhang et al., 2007*), the reconstituted monolayer system is ideal for high resolution structural analy-sis of chemosensory arrays by cryoET for several reasons: 1) the in vitro reconstituted monolayer array is thin (25 nm) and pseudo-crystalline, compared to cells with thicknesses ranging from 500 nm to 1 µm; 2) the monolayer arrays are reconstituted with purified components, hence the system is well-defined, in contrast to native arrays in the crowded cellular environment; 3) the reconstituted system allows for control over which array components are present as well as manipulation of their signaling state; 4) the in vitro system provides large numbers of sub-tomogram volumes (~3000 core-signaling units/tomogram), thereby improving the noise statistics of the sub-tomogram averag-ing process central to achieving a high resolution structure. Using cryoET, we collected and recon-structed, correcting for the contrast transfer function (CTF) of the microscope (*Fernández et al., 2006*), 20 tomograms of monolayers containing reconstituted core-signaling complex arrays.

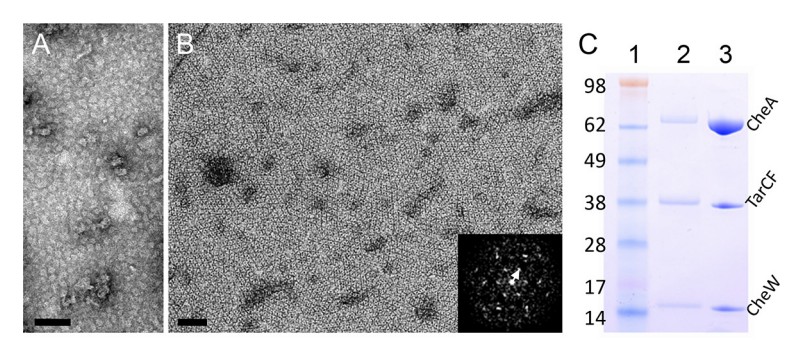

**Figure 1.** Reconstitution of 2D arrays of the receptor signaling complex on lipid monolayers. (**A&B**) Negatively stained electron micrographs of reconstituted lipid monolayers with TarCF only (**A**) or with TarCF/CheA/CheW (**B**). Inset, Fourier transform of a region from the monolayer array, indicating a hexagonal lattice with a 12 nm repeat (white arrow). Scale bars, 100 nm. (**C**) SDS-PAGE gel analysis of the reconstituted monolayer sample (lane 2) and the protein solution (input mixture) used to generate the monolayer arrays (lane 3). Molecular weight markers are indicated (lane 1), and the input mixture contained TarCF:CheA:CheW in a ratio of 9:18:18 µM.

*Figure 2A* (*Video 1*) shows a typical raw tomographic slice (without CTF correction) of a reconstituted monolayer, illustrating patches of 2D lattices with information extending beyond 22 Å (inset, arrow). By extracting and classifying CTF-corrected sub-tomograms, centered on each hexagon of receptor TODs (*Figure 2—figure supplement 1*, yellow circle), we obtained two major classes of the receptor hexagons: one containing a trimer of core-signaling units (CheA$_2$-trimer, *Figure 2B* and *Figure 2—figure supplement 1*, cyan boxes) and one containing a hexamer of core-signaling units (CheA$_2$-hexamer, *Figure 2C* and *Figure 2—figure supplement 1*, orange box). By mapping the individual sub-tomograms from the above two classes onto the original contributing tomograms, we were able to extract the extended lattice organization of the subunits in the monolayer (*Figure 2D*), revealing an interlocking of the CheA$_2$-trimer and CheA$_2$-hexamer classes (*Figure 2E*) consistent with that seen in cellular tomograms (*Briegel et al., 2012*; *2014b*; *Liu et al., 2012*).

To directly compare the lattice organization in our reconstituted monolayer system with that in native *E. coli* cells, we obtained three CTF-corrected tomograms from wt *E. coli* cells that were partially lysed, using a phage-gene-induced instant lysis method that we developed recently (*Fu et al., 2014*), to reduce the sample thickness. Extracting and classifying sub-tomograms containing receptor hexagons from the native *E. coli* cells revealed the same two classes that were observed in the monolayer system. As with the *in vitro* monolayer system, the two receptor hexagon classes observed in native *E. coli* cells also formed an interlocking lattice (*Figure 2—figure supplement 2A*). Extracting sub-tomograms with an extended unit that contained both the CheA$_2$-trimer and CheA$_2$-hexamer, we obtained an average density map for the *in situ* native chemosensory arrays that overlapped very well with the map from the monolayer system (*Figure 2—figure supplement 2B*). Therefore, the *in vitro* reconstituted monolayer system with purified *E. coli* proteins faithfully reproduces the lattice organization found in native *E. coli* cell membranes.

## 3D density maps of CheA$_2$-trimer and CheA$_2$-hexamer

The 3D classification process further improved the resolution of the class-averaged sub-tomograms of CheA$_2$-trimers (*Figure 2B*) to 11.3 Å and CheA$_2$-hexamers (*Figure 2C*) to 17.5 Å resolution, as measured by gold-standard Fourier shell correlation (FSC) (*Figure 3—figure supplement 1A*). A uniform distribution of in-plane orientations of the sub-tomograms and a relatively well sampled, out-of-plane angle enhanced the quality of the averaged density maps (*Figure 3—figure supplement 1B&C*). Nevertheless, some resolution anisotropy exists, with 11 Å in X and Y directions and 15.8 Å in Z direction (*Figure 3—figure supplement 1A*). To take the effect of the anisotropic resolution into account, we low-pass filtered the density map according to the FSC of the Fourier conical shells along various directions (*Diebolder et al., 2015*). The resulting maps of the CheA$_2$-trimer and CheA$_2$-hexamer clearly delineate the density regions corresponding to the receptor, the CheA-P5/CheW ring at the receptor tip, and CheA kinase domain (*Figure 3B&C*, *Figure 3—figure*

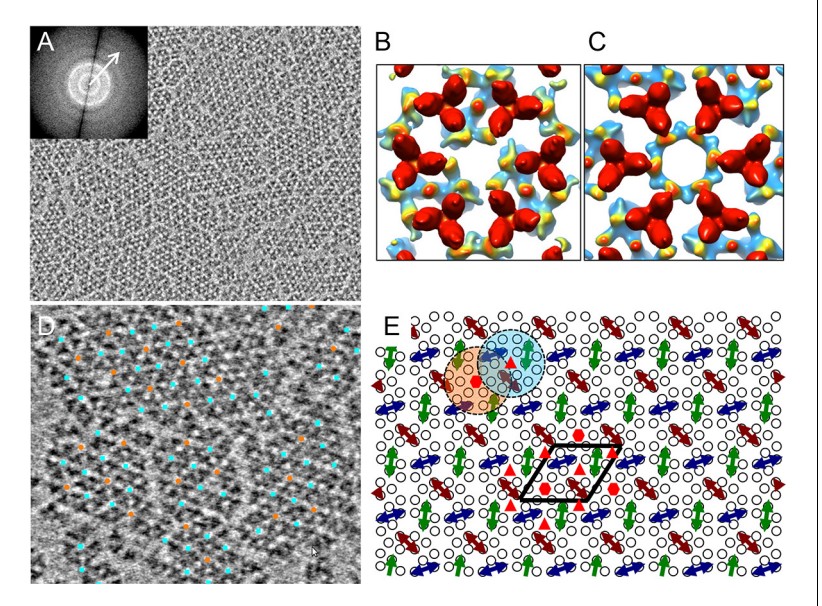

**Figure 2.** CryoET of monolayer arrays of TarCF/CheA/CheW ternary signaling complex. (**A**) A tomographic slice (1.2 nm thick) through the reconstituted monolayer arrays of TarCF/CheA/CheW, without CTF correction. Inset, The Fourier transform of a selected region, displaying Thon rings with information extended to at least 22 Å resolution (arrow). (**B&C**) Averaged density maps of two sub-volume classes containing receptor hexagons (6 TODs) (red), one with a trimer of CheA dimers (CheA$_2$-trimer) (**B**) and the other with a hexamer of CheA dimers (CheA$_2$-hexamer) (**C**). Maps were generated following sub-tomogram volume classification and class-averaging, are contoured at 1.5σ, and are colored according to the height, from the receptor at the top (red) to CheA (blue) below. (**D**) Spatial arrangement of the CheA$_2$-trimer (cyan) and CheA$_2$-hexamer (orange) in the monolayer lattice array, after mapping the classified sub-volumes back onto the tomogram. The array is formed by interlocking CheA$_2$-trimer and CheA$_2$-hexamer subunits. (**E**) A schematic lattice model for the chemosensory arrays. Small circles represent receptor dimers; arrows represent CheA dimers (CheA$_2$). Dashed cyan and orange circles highlight a CheA$_2$-trimer and CheA$_2$-hexamer respectively. The lattice unit cell is outlined in black. Related to *Figure 2—figure supplement 1 and 2*.

The following figure supplements are available for Figure 2:

**Figure supplement 1.** Classification of the sub-tomogram volumes.

**Figure supplement 2.** Comparison of the chemotaxis arrays from native cells and from *in vitro* reconstituted monolayers.

*supplement 2* and further display a number of new features. In particular, the individual receptor dimers are thoroughly resolved, allowing the kinase and CheW/receptor interactions to be isolated to a specific receptor dimer (*Figure 3A–C*). Moreover, the position of the previously unobserved four-helix bundle of the CheA-P3 dimerization domain is clearly discerned to run parallel to the receptor and is positioned close to CheW-interacting receptor dimers (*Figure 3A–C*). In addition, our maps dramatically refine the area of density projecting below the CheA-P5 domain, suggesting that the CheA-P4 kinase domain alone occupies this density region (*Figure 2B, 3A*, and *Video 2*). The CheA-P1 and CheA-P2 domains, on the other hand, are not resolved, likely due to their conformational flexibility.

Regarding the CheA-P5/CheW ring, our density map clearly shows a pseudo three-fold symmetry (*Figure 2B*) in which the density at the CheA-P5/CheW interface between core-signaling units (interface 2) is considerably weaker than the density at the CheA-P5/CheW interface within core-signaling units (interface 1) (*Figure 3A&C*) (*Briegel et al., 2012*). This

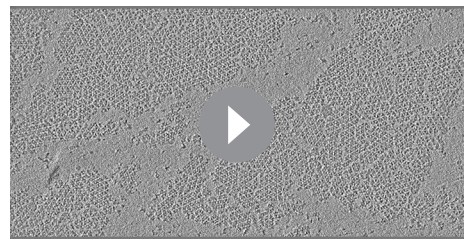

**Video 1.** Tomographic slices of monolayer arrays. Related to *Figure 2*.

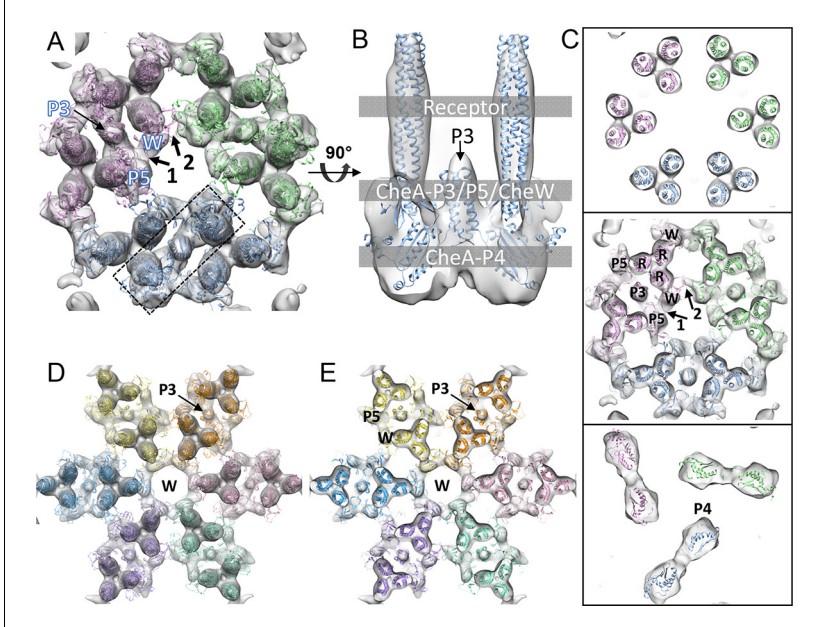

**Figure 3.** CheA$_2$-trimer and CheA$_2$-hexamer density maps with molecular dynamics flexible fitting (MDFF) of computationally constructed *T. maritima* subunit models. (A) Overall fitting of the CheA$_2$-trimer density map contoured at 1.5σ. The three core signaling complexes are colored in pink, blue and green. (B) A sectional view of the boxed region in A, rotated 90˚. The protein components are labeled at the indicated height of the complex (gray boxes). (C) Sectional views of the gray-boxed regions in B at the receptor level (top), the CheA-P3 and P5/CheW ring region (middle), and CheA-P4 region (bottom). (D) Overall fitting of the CheA$_2$-hexamer density map contoured at 1.5σ. (E) A sectional view at the CheA-P3 and CheW-ring region of the CheA$_2$-hexamer density map. In (A-E), CheA-P3, P4, P5, CheW and receptor are labeled as P3, P4, P5, W and R, respectively, and the CheA-P5/CheW interfaces 1 and 2 are indicated. Related to *Figure 3—figure supplement 1*.

The following figure supplements are available for Figure 3:

**Figure supplement 1.** Resolutions of the density maps.

**Figure supplement 2.** X-Z sectional views of the CheA$_2$-trimer density map with MDFF model. The positions of the sections are indicated in the last panel with an orthogonal view (X-Y plane).

**Figure supplement 3.** A metric for the goodness of fit for the docking of the CheA-P4 domain.

finding is in contrast to the previously described pseudo six-fold symmetry of the CheA-P5/CheW ring (*Li and Bayas, 2013*). Most importantly, the previously described 'empty hexagon' that is surrounded by six CheA-occupied hexagons (*Briegel et al., 2012*) is not empty, but rather contains a well-ordered continuous ring of densities (*Figure 2C*) that we were able to unambiguously assign to individual CheW monomers (*Figure 3D&E*). This ring of CheW, as previously speculated (*Liu et al., 2012*), provides additional interactions that couple neighboring receptor TODs and strengthens the interlocking baseplate. Hence, our maps confirm the existence of the CheW ring and establish its participation in the structural foundation responsible for the ultra-stability of the chemosensory array (*Liu et al., 2012*; *Erbse and Falke, 2009*) and for the high cooperativity and extraordinary sensitivity measured in chemotaxis responses (*Goldman et al., 2009*).

## All-atom model of the *T. maritima* chemosensory array

The resolution of our cryoET data permitted the unambiguous assignment of distinct regions of density to specific protein components, enabling the construction of all-atom models of the chemosensory array substructures and extended lattice (*Figure 4—figure supplement 1*). A schematic overview of the modeling procedures carried out in this study is provided in *Figure 4—figure supplement 2* with a more detailed discussion of these procedures located in the Methods section. Briefly, we first constructed models of the receptor TOD, CheA-P3P4 dimer, CheA-P5/CheW ring,

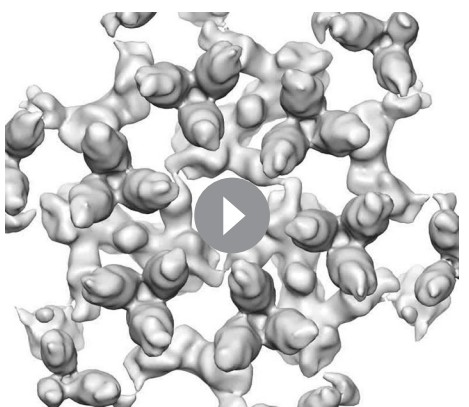

**Video 2.** MDFF model fitting of the CheA2-trimer density map. Related to *Figure 3*.

and CheW-only ring, taking advantage of existing high-resolution X-ray structures from the thermophilic bacterium *Thermotoga maritima* (*Kim and Yokota, 1999*; *Bilwes et al., 1999*; *Park et al., 2006*; *Li and Bayas, 2013*). We then heuristically-arranged, using a 12 nm lattice constant, the resulting component models to produce models of the $CheA_2$-trimer and $CheA_2$-hexamer subunits identified by sub-tomogram classification (*Figure 4—figure supplement 1D&E*). To further refine the key protein-protein interfaces within our atomic models, we adopted a dual MD-based strategy, utilizing both unbiased MD and electron-density-biased molecular dynamics flexible fitting (MDFF) simulations (*Trabuco et al., 2008*). For the subject of our unbiased refinement simulations, we extracted from the $CheA_2$-hexamer model a portion corresponding to the array unit cell, including six receptor TODs, three CheA dimers, and 12 CheW monomers all together arranged as three coupled core-signaling units (*Figure 4—figure supplement 1C*, herein the 'unit-cell model'). For our density-biased refinement simulations, we focused our efforts on the $CheA_2$-trimer model, owing to the higher-resolution of its associated density map, and hence, better resolved MDFF biasing forces. Because the CheA-P4 density is not as well defined as the other parts of the complex, likely due to its conformational flexibility, we carried out a rigid-body docking of the CheA-P4 domain, starting from 10,000 random angular orientations and up to 20 Å shifts from the center of the mass. This fitting exercise resulted in 23 classes separated by 3° and 3 Å (*Figure 3—figure supplement 3A*), generating a metric for the goodness of fit of the P4 domain positioning. In addition to the class of 'best fit'(*Figure 3—figure supplement 3B*, panel 1), one other class, in which P4 is flipped relative to the best fit, was seemingly structurally possible (*Figure 3—figure supplement 3B*, panel 5). However, compared to the best-fit class, this alternative class had a lower cross-correlation value, lower occupancy with only a third the number of contributing fits, and the positions of P4-N and C termini are reversed (flipped), making it hard to connect the P3 and P5 termini with short linkers. Thus, we have focused our efforts and resources on the highest ranking class of P4 position. It should be noted, though, that use of the alternative P4 positioning might produce considerably different MD trajectories. Solvation and ionization of the unit cell and $CheA_2$-trimer models produced systems of size 1.25 million and 1.75 million atoms, respectively, which were subsequently energy minimized and equilibrated for 10 ns, as described in the Methods section. The unit-cell model was then subjected to an 80 ns unconstrained production simulation (*Video 3*), while a 70 ns symmetry-constrained MDFF simulation was used to computationally bias the tertiary structure of the protein components within the $CheA_2$-trimer model according to our 11.3 Å $CheA_2$-trimer density map (*Figure 3*, *Video 2*).

The resulting unit-cell and $CheA_2$-trimer models agreed well with previous structural studies, in particular with respect to the residues participating in the CheA-P5/receptor and CheW/receptor interaction interfaces, as defined by NMR (*Vu et al., 2012*; *Wang et al., 2012*; *Ortega et al., 2013*), crystallography (*Li and Bayas, 2013*), and disulfide mapping studies (*Piasta et al., 2013*; *Natale et al., 2013*). For succinctness, specific residues participating in the various protein-protein interfaces within the array have been listed in *Supplementary file 1*. In addition, the equilibrated model of the *T. maritima* TOD maintained the conserved trimer-forming contacts observed in the X-ray structure of the *E. coli* serine receptor (Tsr) (*Kim and Yokota, 1999*) and revealed two additional trimer-stabilizing salt bridges, namely E387/R389 (conserved as E402/R404 in *E. coli* Tsr) and E351/R403 (structurally homologous to D363/R415 in *E. coli* Tsr) (*Figure 4—figure supplement 3A*). Moreover, in both models, the CheA-P4 kinase domain was seen to stably occupy the region of density directly below the plane defined by the CheW and CheA-P5/CheW rings. Finally, in tandem with the direct visualization of the CheA-P3 dimerization domain in our cryoET density maps, the all-atom model further revealed previously uncharacterized specific interactions between the P3 bundle and

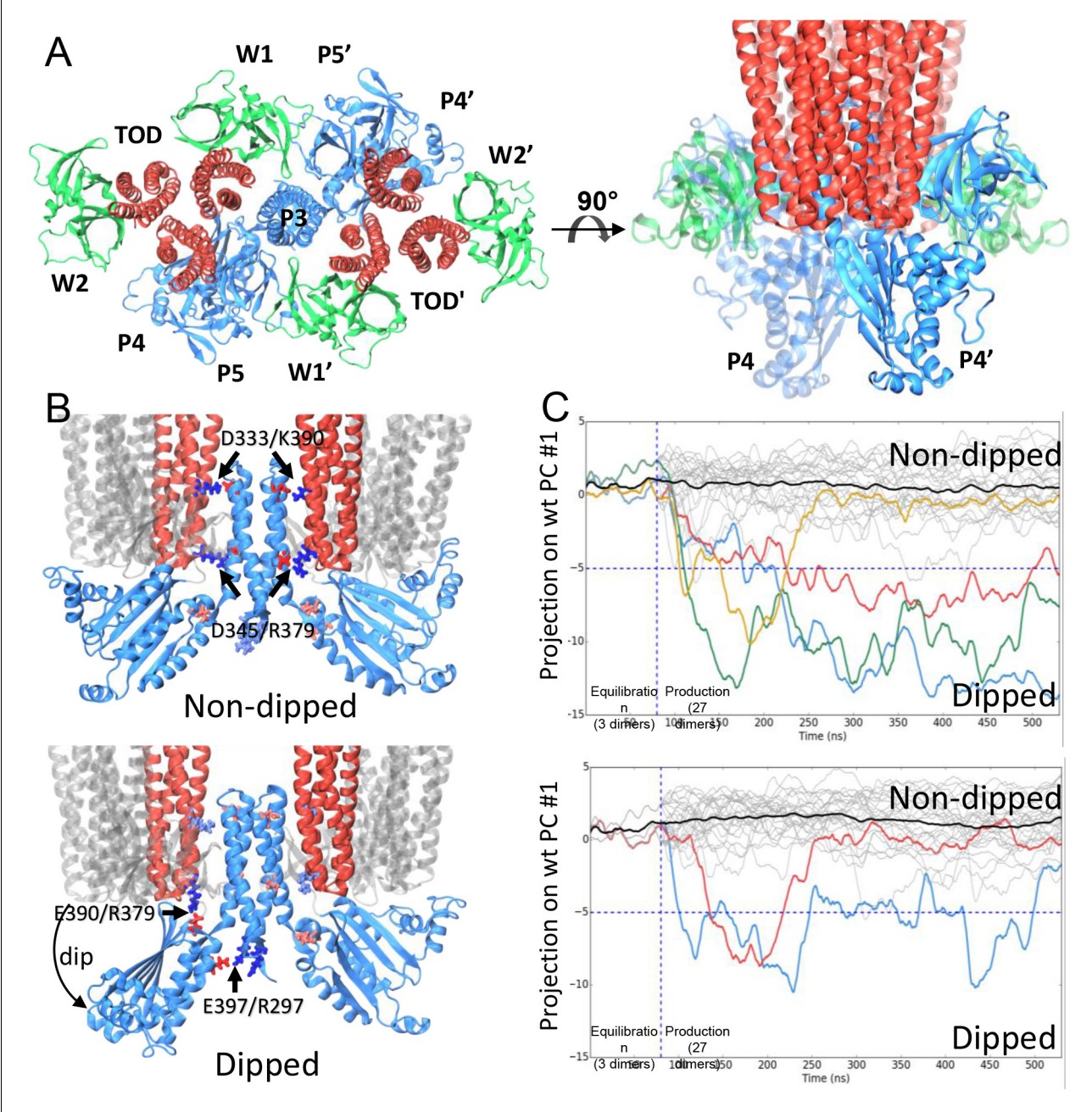

**Figure 4.** CheA dimer conformational switch. (**A**) Top and side views of the core-signaling unit, consisting of two receptor TODs (red), a single CheA dimer (blue), and four CheW monomers (green). (**B**) Two distinct classes, undipped (top) and dipped (bottom), of core-signaling unit structures are present in our MD simulations. Classes differ especially in the orientation of CheA-P4 domain with respect to the rest of the CheA dimer and core signaling unit. Specific contacts that stabilize either conformation are indicated for *T. maritima* and in parentheses for the corresponding residues in *E. coli*. CheA-P5 and CheW have been removed for clarity. (**C**) Time series of CheA dimer conformations extracted from unit cell simulations.. Traces track the projection of the conformations of 27 CheA dimers from the wt (top) and R297A mutant (bottom) unit cell simulations onto the first principal component of the 'dipping' motion. Colored traces track CheA dimers that undergo an extended (>10 ns) 'dipping' motion. Horizontal dashed lines visually demarcate the undipped and dipped CheA dimer classes. Vertical dashed lines separate the initial 80 ns equilibration simulation from nine 450 ns production simulations. Related to **Figure 4—figure supplement 1, 2, 3, 4 and 5**, and **Supplementary file 1**.

The following figure supplements are available for Figure 4:

**Figure supplement 1.** Nomenclatures.

*Figure 4. continued on next page*

*Figure 4. Continued*

**Figure supplement 2.** Overview of molecular modeling and simulation strategy taken in this study.

**Figure supplement 3.** Computational modeling of the extended chemosensory array structure.

**Figure supplement 4.** Overview of key all-atom molecular dynamics simulations conducted in this study.

**Figure supplement 5.** Time series of CheA dimer conformations extracted from CheA$_2$-trimer simulations.

adjacent receptors, involving D333/K390 and D345/R379 contact pairs (I304/N405 and D316/R394 in *E. coli* respectively) (*Figure 4B*, *Supplementary file 1*).

## A conformational change of the CheA kinase domain

The construction of atomic models of the array unit cell and subunits permitted the use of equilibrium all-atom MD simulations to further investigate the molecular details of dynamic events potentially relevant to signaling. An overview of the key MD simulations conducted in this study is given in *Figure 4—figure supplement 4A*. The unit cell system contains the minimal set of components needed to represent the full receptor signaling array, which was made possible through the use of periodic boundary conditions to mimic the bulk symmetry of the chemosensory array, preventing the need to interpret potentially problematic effects due to unconstrained boundaries (*Figure 4—figure supplement 2C*, *Video 3*). We conducted a series of nine simulations of 450 ns each, using the equilibrated unit-cell model; additionally, we ran ten, 120 ns simulations of the equilibrated CheA$_2$-trimer system for comparison with the unit-cell simulations. Intriguingly, our simulations of both models revealed an ensemble of distinct core-signaling unit conformations (*Figure 4 B&C*), including structures in which the associated CheA dimer displayed either an undipped conformation (*Figure 4B*, top) or dipped conformation (*Figure 4B*, bottom). In the latter case, the P4 domain of one CheA monomer adopted a 'dipped' state through rotations about the P3-P4 and P4-P5 flexible linkers, significantly affecting its contacts with neighboring receptor dimers and the P5 domain (*Video 4*). As many biochemical, biophysical, and mutational studies have implicated dynamic structural changes within these regions of the core-signaling unit during the propagation of signals (*Piasta et al., 2013*; *Natale et al., 2013*; *Wang et al., 2014*; *Briegel et al., 2013*), we systematically identified the distinct structural classes of core-signaling unit conformations present in our MD simulations and isolated them for comparative analysis. Specifically, we used the UPGMC hierarchical clustering method (*Müllner, 2013*) to assign the conformations of the 27 core-signaling units sampled in our unit cell simulations (3 units/unit cell) to groups of similar structure based on their pairwise root-mean-square deviation (RMSD). Cross-examination of structures within the resulting core-signaling unit clusters revealed the formation of two new salt bridges stabilizing the 'dipped' state, namely R297/E397 (R265/E368 in *E. coli*) between the P3 and P4 domains and E390/R379 (E361/R394 in *E. coli*) between the P4 domain and nearby receptor tip (*Figure 4B*, bottom). Moreover, to accommodate the reorientation of the P4 domain, the P3 dimerization bundle was observed to break the receptor contacts (D333/K390 and D345/R379) observed in the 'undipped' state (*Figure 4B*, *Video 4*), suggesting that the mobility of the P3 bundle plays a key role in the conformational dynamics of the CheA dimer.

We next sought to examine the temporal evolution of the dipping motion in each of the CheA dimers present in our simulations. For this purpose, we used Principal Component Analysis (PCA) to systematically derive, from the trajectory of a single dipping CheA dimer, a pseudo reaction coordinate by which to easily monitor the progression towards the 'dipped' conformation. A total of four 'dipping' events were observed in our unit cell simulations, as illustrated by projection of the conformations of the 27 CheA dimer time series onto the first principal component (*Figure 4C*, top). Importantly, an additional two dipping events were observed in the 30 CheA dimers of the relatively shorter simulations of CheA$_2$-trimer model (*Figure 4—figure supplement 5*), demonstrating that the ability of the conformational change to occur is not an artifact of the particular choice of CheA P4 positioning during modeling. Interestingly, the three *extended* 'dipping' events observed in the unit-cell simulations (*Figure 4C*; red, blue, and green traces) as well as the two events observed in

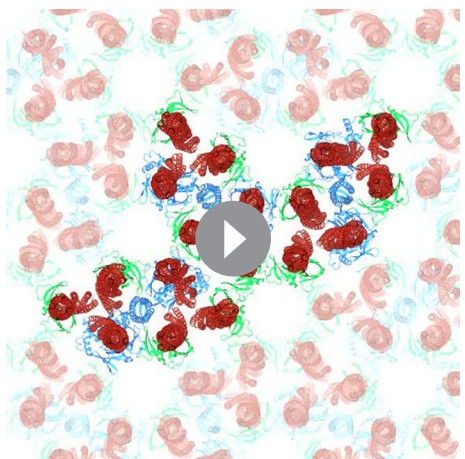

**Video 3.** Molecular dynamics simulation of array unit cell. Shown here is a 75 ns clip of a wild type unit cell trajectory, illustrating the dynamics of the 1.2 million atom model, including 6 receptor TODs (red), 3 CheA dimers (blue), and 12 CheW monomers (green). Periodic images, shown here with reduced opacity, enforce the boundary conditions of the extended array architecture but are not simulated explicitly. Solvent and ions have been removed for clarity. Related to *Figure 4*.

the CheA$_2$-trimer simulations were accompanied by the formation of the R297/E397 contact. Notably, this contact was not formed in the one *short* dipping event, which returned to the 'undipped' bulk state (*Figure 4C*; gold trace), suggesting that the R297/E397 contact may play a role in stabilizing the 'dipped' state. To further investigate the significance of the R297/E397 contact for the conformational dynamics of CheA, we launched nine additional unit cell simulations with an R297A mutation to prevent the potential formation of the R297/E397 salt bridge. Indeed, while two CheA dimers exhibited the dipping motion in these simulations, including one dimer that underwent two dips, the mutants quickly return to the bulk (*Figure 4C*, bottom).

## Biochemical validation of CheA conformational change in *E. coli* cells

To determine if the CheA-P4 dipping motion observed in the MD simulations of the *T. maritima* chemosensory array is sampled in the native chemotactic response of *E. coli*, we carried out cysteine disulfide cross-linking experiments. In particular, we tested the interaction state (I304/N405 and D316/R394) or only in the

interface for contacts existing in the undipped dipped state (E361/R394) (*Figure 5B*). Notably, in the simulations, R394 of Tsr switches its contact with D316 of CheA-P3 to E361 of CheA-P4 during the transition of the CheA dimer from 'undipped' to 'dipped' (*Video 4*).

Using soft-agar assays, it was seen that the chemotactic ability of the I304C/N405C double cysteine mutant is appreciably compromised compared to that of the control (cysless CheA/ wt Tsr), any of the single mutants (I304C/wt Tsr, cysless CheA/N405C, cysless CheA/N405S), and when one half of the pair has been mutated to serine (I304C/N405S) (*Figure 5A*), suggesting that dynamic interaction between CheA-P3 and the receptor is important for chemotactic function. Moreover, in vivo cross-linking and western blot analysis showed a high molecular weight band present only in the double cysteine mutant, suggesting the presence of species formed by cross-linking between CheA-P3 and Tsr (*Figure 5B*). We also examined cross-linking residue pairs that involve Tsr-R394 interactions with CheA, one in the 'undipped' state (CheA-E316C/Tsr-R394) and the other in the 'dipped' state (CheA-E361C/Tsr-R394). When Tsr-R394 is replaced by a cysteine or serine, either as a single mutant (cysless CheA/R394C, cysless CheA/ R394S) or in the context of a double mutant

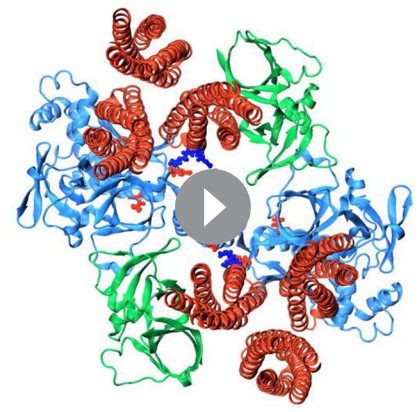

**Video 4.** Molecular dynamics simulations reveal conformational switch in CheA P4 domain. Shown here is one of four 'dipping' events observed in the wild type unit cell simulations, leading to modified contacts between the CheA dimer and receptor TODs. Strong contacts between P3 and neighboring receptor dimers (D333/K390 shown here with licorice representation) are disrupted in favor of new contacts between P3/P4 and P4/receptor stabilizing the dipped state (R297/E397 and E390/R379 respectively, shown here with licorice representation). Related to *Figure 4*.

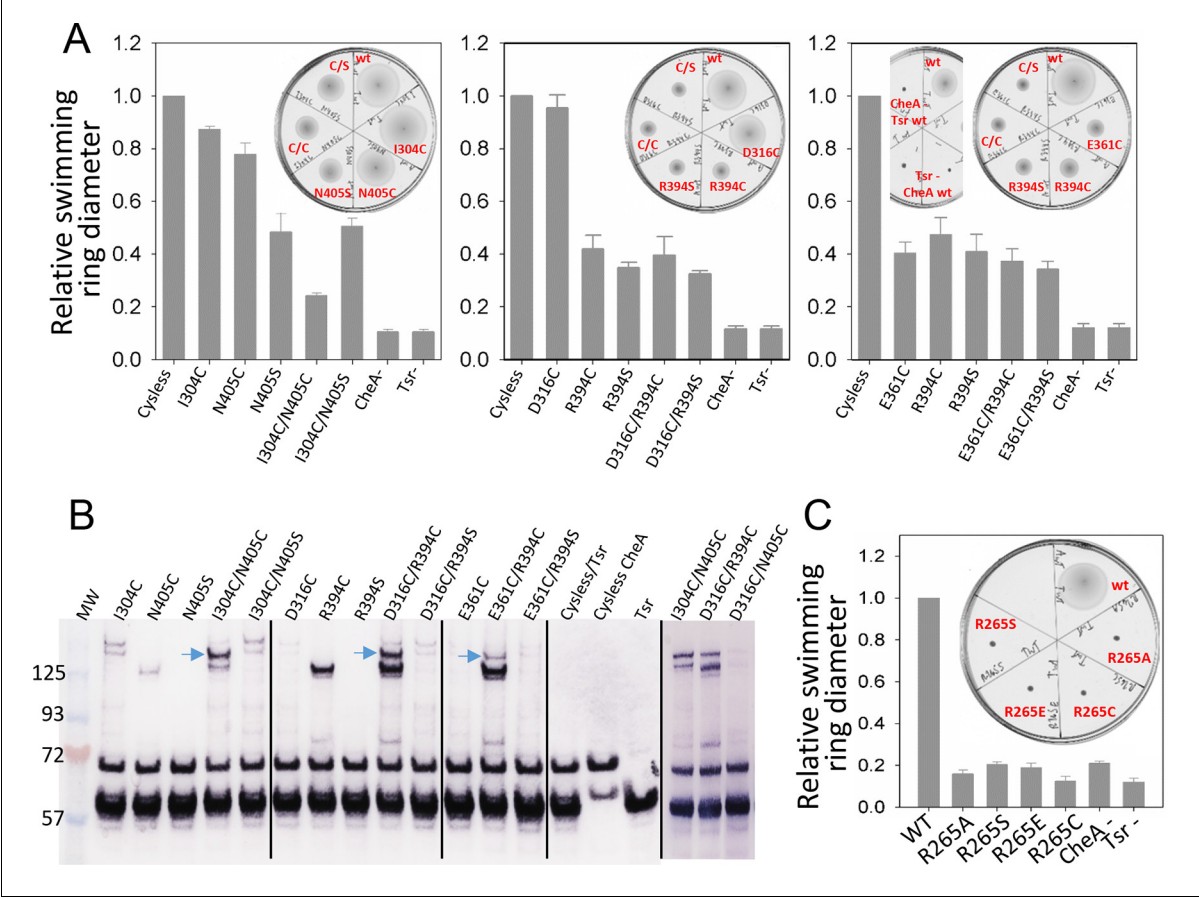

**Figure 5.** Biochemical validation of alternative CheA conformations in *E. coli*. (**A**) Swimming ability of *E. coli* cells with mutations in the CheA-P3 and Tsr interface (I304/N405 and D316/R394) and in the 'dipped' CheA-P4 and Tsr interface (E361/R394). Swimming activities are normalized to the cysless CheA and wt Tsr, ± standard deviation (*n*=6). Inset, representative images of soft agar plates for swimming ability, with specific constructs labeled in red. (**B**) Disulphide cross-linking of the CheA-P3 and Tsr interface (I304C/N405C and D316C/R394C) in the undipped CheA dimer conformation (top panel of *Figure 5B*) and the CheA-P4 and Tsr interface (E361C/R394C) occurring in the dipped CheA-P4 'dipped' conformation (bottom panel of *Figure 5B*). Non-reducing (top) and reducing (bottom) SDS-PAGE gels were analyzed by immunoblotting for Tsr and CheA. Cross-linked species were indicated with blue arrows. (**C**) Swimming ability of *E. coli* cells with mutations at R265 of CheA-P3 domain, normalized to the wt, ± standard deviation (*n*=8). Related to *Figure 5—figure supplement 1*.

The following figure supplements are available for Figure 5:

**Figure supplement 1.** CryoEM images of plunge-frozen *E. coli* cells expressing WT Tsr and WT CheA (**A&B**), R265A CheA (**C**), R265C CheA (**D**), R265S CheA (**E**), and R265E CheA (**F**). The arrays are marked with white curved arrows. Scale bars, 100 nm.

(D316C/R394C, D316C/R394S, E361C/R394C, E361C/R394S), the chemotaxis function of *E. coli* is partially inhibited. On the other hand, the chemotactic ability of CheA-E361C as a single mutant (E361C/wt Tsr) is also partially inhibited, while CheA-D316C (E316C/wt Tsr) mutation bears no effect on the function (*Figure 5A*). Furthermore, the cross-linking pattern of both Tsr-R394 mutant pairs showed two high molecular weight bands corresponding to distinct cross-linked species, one with a disulfide formed between Tsr and CheA (*Figure 5B*, upper band, blue arrows) and the other with a disulfide between two Tsr molecules with the R394C mutation (lower band). Interestingly, the cross-linking of CheA-P4/Tsr (E361C/R394C) in the predicted 'dipped' conformation is much weaker than the cross-linking of CheA-P3/Tsr (D316C/R394C) in the 'undipped' state, though both involve the same R394 residue of Tsr (*Video 4*). The lower cross-linking efficiency could be due to the relatively infrequent occurrence of the CheA 'dipped' conformation, and/or because the residues are further apart in a dominant conformation, suggesting that the CheA-P4 'dipped' conformation observed in silico may have been sampled within the native chemosensory complex of *E. coli*.

Our MD simulations of the *T. maritima* unit cell further indicated that R297 on the CheA-P3 domain is potentially involved in the stabilization of the conformational transition of the CheA-P4 (*Figure 4C*, *Video 4*). Indeed, substitution of the corresponding residue in *E. coli* (R265) with several amino acids of different properties (R265C/S/A/E) were all detrimental to the chemotactic function of *E. coli* as measured by the soft-agar assay, without affecting the cluster formation (*Figure 5C*, *Figure 5—figure supplement 1*). Since this residue is located at the N-terminus of the four-helix P3 dimerization motif, R265 could direct the P2-P3 linker away from the cis subunit and toward the trans subunit, thus anchoring CheA-P1P2 to CheA-P4' for trans-interaction and phosphorylation (*Bilwes et al., 1999*). A more complete model of the core-signaling complex for *E. coli* may be necessary to fully interpret the drastic impact of this single CheA residue on the entire chemotactic machinery.

## Discussion

While much effort has been expended in the derivation of models to describe the transduction of ligand-binding events within the receptor proteins, including an established piston mechanism and a hypothesized alternating static–dynamic 'yin-yang' on-off switch model (*Falke and Piasta, 2014*), how the structure and dynamics of the CheA kinase are affected during signaling remains poorly understood. In this study, we identified, using MD simulations, a dipping motion of the CheA P4 domain, which was functionally characterized using swim assay and cross-linking experiments. While the role of the predicted conformational change in CheA is not immediately clarified in the preliminary biochemical experiments carried out here, our model highlights the importance of CheA dynamics for signaling and suggests that the dynamics of the P4 kinase domain, in particular, warrants special investigation. More importantly, the atomic model presented here, in general, provides improved knowledge of the positioning of the P3 and P4 domains, incorporates the presence of the CheW only ring, and identifies probable novel side-chain contacts within the extended chemosensory architecture. Further improving the resolution of our cryoET data to better than 8 Å using the novel lipid-monolayer system described above would allow generation of an atomic homology model of the *E. coli* chemosensory array, greatly facilitating the use of the wealth of existing biochemical and biophysical data and providing directly transferrable structural and dynamical predictions. We hope that the findings presented here will inspire further experimental and computational studies towards the elucidation of a complete mechanistic description of signal transduction and amplification within this truly impressive biological sensory apparatus.

## Materials and methods

### Materials

Plasmids and cell strains used in this study were gifts from Dr. Parkinson, University of Utah, except for plasmid pHTCF (kind gift from Dr. Weis, University of Massachusetts, Amherst). Plasmid pHTCF is an IPTG-inducible expression vector for the N-terminal His$_6$-tagged cytoplasmic fragment of wt aspartate receptor (TarCF) that contains residues 257–553. Plasmids pKJ9 and PPA770 are IPTG-inducible for the expression of CheA and CheW, respectively. Plasmid pRR53 is an IPTG-inducible expression vector (amp$^R$) for the wt serine receptor (Tsr). Plasmid pGP26 is a sodium salicylate (Na-S)-inducible expression vector (cam$^R$) for cysteine-less CheA and wt CheW. Plasmids pRR53 and pGP26 were used to generate mutations in Tsr and CheA, respectively.

### Protein expression and purification

*E. coli* strain RP3098, which lacks all Che proteins and chemoreceptors, was transformed with plasmid pKJ9 or PPA770 for CheA or CheW expression, respectively. CheA expression was induced at an OD$_{600}$ of 0.6–0.8, with 1 mM IPTG, overnight at 15°C. CheA was purified using an Affi-gel Blue column (Bio Rad, Hercules, CA) followed by gel filtration on a Superdex 200 column. Further purification with a Mono Q ion exchange column resulted in >99% homogeneity with an overall yield of 50 mg/L of cells. CheW expression was induced by the addition of IPTG (0.5 mM), at an OD$_{600}$ of 0.4–0.6, at 37°C. CheW was purified through 20%–40% ammonium sulfate precipitation, a DEAE column followed by a MonoQ ion exchange column and a Superdex 75 size exclusion column. This

procedure resulted in highly purified CheW with a yield of 6 mg/L of cells. His$_6$-tagged wt TarCF (His$_6$-TarCF$_{QEQE}$) was expressed in DH5alpha cells with plasmid pHTCF. TarCF was induced by the addition of IPTG (0.5 mM) at an OD$_{600}$ of 0.4–0.6 at 37$^\circ$C and purified with a Ni$^{2+}$-NTA affinity column followed with a mono Q column for quick removal of imidazole, without dialyzing overnight. The yield for TarCF was excellent (120 mg/L of cells).

## Monolayer reconstitution

A Ni$^{2+}$ lipid containing monolayer system was used to reconstitute the chemotaxis core-signaling complex arrays. A mixture of 9:18:18 µM of TarCF:CheA:CheW in a buffer containing 75 mM Tris-HCl, pH 7.4, 100 mM KCl, 5 mM MgCl$_2$ was applied to a Teflon well, over which we immediately lay a lipid monolayer containing 2:1 DOPC:DOGS-NTA-Ni$^{2+}$ lipid mixture, at 2 mg/ml concentration. The monolayer set up was left undisturbed in a humidity chamber overnight. The monolayer specimen was picked up with holey carbon grids, stained with 1% uranyl acetate, and examined with an FEI T12 microscope operated at 120 KV.

## Cryo-electron tomography

Reconstituted monolayers using the best conditions identified by negative staining (*Figure 1B*), were picked up with perforated R2/2 Quantifoil grids (Quantifoil Micro Tools, Jena, Germany) pre-coated with 10 nm fiducial gold beads on the backside of the grid and plunge-frozen using a manual gravity plunger. This method prevents disruption of the monolayer by supporting single-side blotting which eliminates the contact between the blotting filter paper and the delicate monolayer. The frozen-hydrated EM grids were loaded into FEI Polara cartridges and imaged under low-dose conditions using a Tecnai Polara microscope (FEI Corp., OR.) operating at 200kv. A series of low dose projection images were recorded with tilt angles ranging from 70$^\circ$ to -70$^\circ$ with a Gatan 4K × 4K CCD camera (Gatan, Inc., PA), at a nominal magnification of 39,000×, with a defocus value of 5–8 µm and an accumulated dose of ~60 e$^-$/Å$^2$. A total of 32 tomographic tilt series were collected using an FEI automated tomography software.

## 3D reconstruction, sub-tomogram classification and averaging

Of the 32 tilt series collected, 20 tilt series with negligible mechanical or physical artifacts were selected for image processing and tomographic volume reconstruction. The monolayer produces an ideal EM specimen: it is thin (25 nm) and also provides strong signals in power spectra, due to near-crystalline packing of the protein components (*Figure 2A* inset), allowing accurate determination of the Contrast Transfer Function (CTF) using strip-based periodogram averaging in TomoCTF (*Fernández et al., 2006*). The tilted projection series were roughly aligned using IMOD (*Kremer et al., 1996*), and the alignment parameters were further refined using fiducial-free Area Matching with Geometry Refinement as implemented in Protomo (*Winkler, 2007*). Using the refined geometry parameters, the raw projections were centered and rotated so the tilt azimuth was coincident with the Y-axis using the IMOD 'newstack'function. These rotated stacks were corrected for the CTF with phase flipping, and volume reconstructions were made using SIRT as implemented in IMOD. These were calculated using a GPU, thereby removing an additional interpolation in the reconstruction step, by avoiding the use of cosine stretching of the input projections. Reconstructed volumes calculated from 20 SIRT iterations, providing higher contrast, were used for the initial cycles of sub-tomogram extraction and alignment, while those from 60 SIRT iterations were used for the final cycles.

To extract sub-tomograms, initial positions of the receptor complexes, respective to a Cartesian grid defined by each tomogram, were approximated by using a template matching algorithm implemented in Matlab with a reference that emphasized the receptor dimers with little influence from CheA. Both the template and tomograms were low-pass filtered to 4 nm and binned by 3. This resolution, as well as a coarse angular search, were chosen to eliminate any statistical correlation of high resolution information between half data sets in later image processing steps. Following template matching sub-volume extraction, the data were randomly segregated into two groups, which were processed independently for all subsequent steps.

Sub-tomogram alignment and classification were carried out using Protomo's i3 image processing utilities (*Winkler, 2007*). Using Multivariate Statistical Analysis and Hierarchical Ascendant

Classification, eight class averages were produced from each half data set by focusing the analysis on the CheA portion of the complex. Initial references for each half set were generated by choosing averages from eight classes. These references were then used to align class averages chosen to each have ~50 contributing sub-volumes. In the following cycle, the raw sub-tomograms were subject to multi-reference alignment, but only a small in-plane and translational adjustment was allowed. This alignment by classification was repeated five times, while allowing the automatic exclusion of high variance outliers after the second cycle. In addition to the $CheA_2$-trimer and $CheA_2$-hexamer classes (*Figure 2B, C*), divergent organizations of CheA/receptor complex were also included as references (*Figure 2—figure supplement 1*). After the final cycle, class averages containing either $CheA_2$-trimer or $CheA_2$-hexamer were manually selected and averaged together for each half data set, and the corresponding gold-standard FSC was calculated to evaluate the reliability of the data. Soft cylindrical masks were used, rather than spherical masks, given the extended slab like nature of the specimen. The final averages of $CheA_2$-trimer or $CheA_2$-hexamer from two half data sets of 3,000 sub-volumes or 300 sub-volumes, respectively, were combined and an empirical correction for the CTF envelope was applied for sharpening, which helped to further clarify the receptor dimers, as well as the P3 dimerization domain.

To access the degree of resolution anisotropy, conical Fourier shell correlations from the two independent half data sets of $CheA_2$-trimer, along each of the principal axes, as well as the 10 axes bisecting them, were calculated (*Diebolder et al., 2015*). The averaged density map of $CheA_2$-trimer was then low-pass filtered according the conical FSCs along three principle axes by using cones with a 42° half-angle, adjusted for any overlapping regions in reciprocal space.

## Computational modeling of *T. maritima* core components: Receptor trimer-of-dimers (TOD)

A model of the cytoplasmic portion of the *T. maritima* receptor dimer was taken from the X-ray crystal structure of the TM1143 chemoreceptor (PDB 2CH7) (*Park et al., 2006*). Using the *E. coli* receptor TOD (PDB 1QU7) (*Kim and Yokota, 1999*) as a reference, a *T. maritima* receptor TOD model (*Figure 4—figure supplement 1A*, *Figure 4—figure supplement 2A*) was obtained by arranging individual receptor dimer models from the previous step so that homologous trimer-forming contacts were preserved. *CheA-P34:* An atomic model of the soluble *T. maritima* CheA dimer, including the dimerization (P3) and kinase (P4) domains, was based on atomic coordinates from the X-ray crystal structure PDB 1B3Q (*Bilwes et al., 1999*). *CheA-P5/CheW and CheW rings:* Atomic models for both the CheA-P5/CheW and CheW rings were based on the X-ray crystal structure of the Receptor/CheA-P5/CheW ternary complex, PDB 4JPB (*Li and Bayas, 2013*). In the case of the CheW ring model, the P5 domains of the CheA-P5/CheW ring model were exchanged with CheW monomers, using the dual-SH3-like fold shared between by CheA-P5 and CheW, to obtain an appropriate placement and orientation with respect to the neighboring monomers. *Figure 4—figure supplement 3* schematically summarizes the modeling procedures described above. All missing loops were added using MODELLER (*Sali, 1993*). The TOD, CheA-P5/CheW, and CheW ring core component models were subjected to 150 ns of equilibration to ensure their structural integrity.

## Construction of *T. maritima* array subunit models

The $CheA_2$-trimer and $CheA_2$-hexamer subunits models (*Figure 4—figure supplement 1D&E*) were constructed heuristically; using as a visual reference the extended organization of kinase-filled and kinase-empty rings evident in our density maps to arrange the components, also assuming an approximate 12 nm lattice constant (*Figure 4—figure supplement 3B*). Next, we made use of the CheA-P5/receptor interface from the ternary complex structure PDB 4JPB (*Li and Bayas, 2013*) to model the CheW/receptor interface, assuming a receptor-binding mode homologous to that of CheA-P5. Using the CheA-P5 and CheW monomer/receptor models from the previous step, positional constraints on the receptor TODs were set relative to the height and orientation of the protein rings. Finally, CheA-P3,4 core component models were placed between adjacent TODs in accordance with the patterns observed in our density maps and joined to nearby ring-bound regulatory domains (P5) at the P4-P5 flexible linker. From the $CheA_2$-hexamer model we then extracted a portion corresponding to the array unit cell (*Figure 4—figure supplement 2C*, *Figure 4—figure supplement 3C*) for further study with all-atom MD simulations. In addition, symmetry-constrained

molecular dynamics flexible fitting (MDFF) simulations (*Trabuco et al., 2008*) were used to refine the overlap between our experimental densities and heuristically constructed $CheA_2$-trimer and $CheA_2$-hexamer subunit models (*Figure 4—figure supplement 3D, Video 2*). The Situs modeling package (*Wriggers, 2010*), was used to rigidly dock the subunit models into their respective cryoET maps to provide the initial overlap for our MDFF simulations.

## Molecular dynamics simulations

The array unit cell model was hydrated with TIP3P water molecules using VMD's solvate plugin (*Humphrey et al., 1996*), producing a simulation box defined by hexagonal lattice parameters a=208 Å, b=208 Å, c=334 Å, $\alpha$=90°, $\beta$=90°, $\gamma$=120°. Using VMD's autoionize plugin, the hydrated system was then neutralized and subsequently ionized with sodium and chloride ions to the physiological concentration of 150 mM, resulting in a model containing 1,153,756 atoms. The unit cell model was then subjected to a series of conjugant gradient energy minimizations (300,000 steps in total) and restrained NPT equilibration simulations (10 ns in total). In the same fashion, the $CheA_2$-trimer and $CheA_2$-hexamer subunit models were hydrated and ionized to produce systems of size 1,751,375 atoms (245x245x310 Å) and 4,588,588 atoms (385x405x310 Å) respectively. Each subunit model was then subjected to the same minimization (300,000 steps) and restrained NPT equilibration (10 ns) scheme as the unit cell model. An outline of subsequent equilibration and production simulations is given in *Figure 4—figure supplement 4A*. Production simulations of the unit cell and MDFF-refined $CheA_2$-trimer models were conducted with weak (spring constant = 0.1 kcal/mol*nm$^2$) harmonic restraints placed on the alpha carbons of the first five membrane-proximal receptor residues to maintain TOD splay in the absence of membrane and crowding agents. In the case of the post-MDFF production simulations of the $CheA_2$-trimer, additional weak harmonic constraints were placed on the outermost CheW and CheA-P5 domains to enforce the bulk array boundary conditions, as the trimer organization does not permit the use of periodic boundary conditions to represent the necessary symmetry.

All molecular dynamics simulations were performed using the parallel molecular dynamics code, NAMD 2.9 (*Phillips et al., 2005*) and CHARMM22 force field (*MacKerell et al., 1998*) with CMAP corrections (*Mackerell et al., 2004*). Equilibrium simulations were conducted in the NPT ensemble with isobaric and isothermal conditions maintained at 1 atm and 330 K for equilibration, or 350 K for production using the Nosé-Hoover Langevin piston, with a period 200 femtoseconds (fs) and relaxation time of 50 fs, and the Langevin thermostat with a temperature coupling of 5 ps$^{-1}$. The r-RESPA integrator scheme with an integration time step of 2 fs was used (*Phillips et al., 2005*). SHAKE constraints were applied to all hydrogen atoms. Short-range, non-bonded interactions were calculated every 2 fs with a cutoff of 12 Å and long-range electrostatics were evaluated every 6 fs using the particle-mesh-Ewald (PME) method with a grid size of 1 Å. Periodic boundary conditions with fixed cross-sectional area (x-y plane) were used. MDFF simulations were performed in the NVT ensemble at 330 K using the settings described above with additional restraints applied to prevent loss of secondary structure, chirality errors, and the formation of cis-peptide bonds.

## Simulation analysis

Visualization and extraction of raw trajectory data for analysis were performed using VMD. Principal Components Analysis (PCA) was carried out using custom scripts (source code file *PCA.py*) involving the Numpy, Scipy, and MDAnalysis python packages (*Michaud-Agrawal et al., 2011*). For the PCA analysis, a single dip-exhibiting CheA dimer was isolated from one of our wild-type unit cell simulations, and each frame (23,331 frames in total) was aligned to the initial CheA dimer model using the P5 domains (residues 543–671). Principal components were computed using the alpha carbons of the P4 domains (residues 352 to 542). The fractional variances accounted for by the top three modes were 41.8%, 31.1%, and 8.1% respectively. Subsequently, the three CheA dimers from each replica of the wild-type unit cell model (27 dimers total), R297A unit cell model (27 dimers total), and $CheA_2$-trimer model (30 dimers total) simulations were extracted, aligned to the P5 domains, and projected on to the top principal component of the wild-type dip-exhibiting CheA dimer. These projections were grouped according to model type to create *Figure 4C* (top and bottom) and *Figure 4—figure supplement 5*. Illustrations of the PCA results were produced using the python-plotting package, Matplotlib. Clustering analysis was performed using custom scripts (source code

file *clustering.py*) involving the python packages noted above as well as the implementation of the UPGMC hierarchical, agglomerative clustering algorithm from the fastcluster package (*Müllner, 2013*). For the clustering analysis, we first extracted the three core-signaling units from each of the nine wild-type unit cell replica simulations, using 1500 frames/core-signaling unit for a total of 40,500 frames. The RMSD distance matrix was then computed using the 'rms_fit_trj' function from the MDAnalysis package. Our analysis identified four major clusters of structures within the above distance matrix with relative populations of 80%, 10%, 10% and 2%, representing the undipped and dipped CheA dimer states as well as two intermediate states respectively.

## Mutagenesis

Specific mutations on CheA and Tsr were generated by site-directed mutagenesis on the background of cysteine-less CheA (pGP26) (*Miller et al., 2006*; *Zhao and Parkinson, 2006*) and wt Tsr (pRR53), respectively. Each mutation was introduced using a pair of primers (Integrated DNA Technologies, Inc., Coralville, Iowa), complementary to the template except for the site of mutation, and PfuUltra II Fusion HS DNA polymerase (Agilent Technologies, Santa Clara, CA ), following the manufacturer's thermocycling parameters. The presence of the mutations was confirmed by DNA sequencing.

## Cross-linking and western blot analysis

Starter cultures were grown in LB broth (10% tryptone/5% yeast extract/10% NaCl), supplemented with appropriate antibiotics, overnight at 37°C with 250-rpm shaking. Subsequently, the overnight cultures were diluted 1:50 into a 5-ml LB broth, supplemented with the appropriate antibiotics and allowed to grow at 37°C with 250-rpm shaking. When the optical density at 600 nm of the cultures reached ~0.8, cells were induced with 100 μM IPTG and 0.6 μM Na-S, in the presence of 100 μM serine, for 1 hr at 37°C. After induction, cells were collected by centrifugation (3000 x g, 4°C for 10 mins) and then re-suspended in cold PBS, in the presence of 100 μM serine. Cross-linking was initiated by addition of 60 μM copper (II) sulfate and 200 μM phenanthroline (1 hr, RT) and stopped by addition of 20 mM iodoacetamide and 3.7 μM neocuproin. Cells were immediately mixed with 4× NuPAGE lithium dodecyl sulfate/PAGE sample buffer (Invitrogen Corp., Carlsbad, CA), with or without reducing agent (dithiothreitol), and then boiled for 5 min before electrophoresis. Samples were analyzed on 4–12% SDS-PAGE gels in MES running buffer ((Invitrogen Corp., Carlsbad, CA)). Gels were transferred to nitrocellulose membranes, blocked, and immunoblotted by using antiserum against Tsr (1:2500) and CheA (1:1250) (gifts from Dr. Subramaniam, NIH), followed by an alkaline phosphatase conjugated anti-rabbit antibody (1:50,000, Sigma). Bands were detected on the membrane using an NBT/BCIP kit (Promega Corporation, Madison, WI)following the manufacturer's instructions.

## Soft agar assays

The UU2682 strain does not express any of the chemoreceptors, CheA or CheW, rendering it non-chemotactic despite the presence of an intact flagellar system. Presence of both pRR53 (wt-Tsr) and pGP26 (cysless CheA and wt-cheW) is required to rescue the chemotaxis of UU2682, observed as formation of attractant rings on a soft-agar media. To assess the effect of mutations on Tsr and/or CheA on the chemotactic ability of *E. coli*, the mutant plasmids were introduced into the UU2682 strain and assayed for formation of attractant rings. The soft agar assay protocol used here is adapted from the Parkinson laboratory (University of Utah). Fresh colonies were plated on LB-agar media (10% tryptone/5% yeast extract/10% NaCl/10% agar), supplemented with carbenicillin (100 μg/ml) and chloramphenicol (34 μg/ml), and grown overnight at 37°C. Next day, using a fine-tip toothpick, colonies were picked from the fresh LB-agar plate and stabbed into a soft-agar media (10% Tryptone/5% yeast extract/5%NaCl/0.27% agar) containing antibiotics (carbenicillin 50 μg/ml, chloramphenicol 17 μg/ml), inducers (100 μM IPTG and 0.6 μM Na-S) and 100 μM serine. Plates were then incubated at 32°C for ~8 hr and the diameter of attractant rings immediately measured after incubation.

## Acknowledgments

We thank Drs. JS Parkinson for bacterial strains and plasmids and S Subramaniam for antisera against chemotaxis components, D Bevan and P Greer for computer technical support and A Makhov for cryoEM instrumental support. In addition, we thank Drs. Wei Han and Davi Ortega for insightful discussions regarding the computational aspects of this study. We also thank Dr. T Brosenitsch for reading the manuscript. This work was supported by the National Institutes of Health NIGMS Grant R01GM085043 (PZ), P50GM082251-7518 (PZ), 9P41GM104601 (KS), and 5R01GM098243 (KS) as well as the National Science Foundation PHY-1430124 (KS). Large-scale molecular dynamics simulations were performed on the Blue Waters supercomputer, financed by the National Science Foundation (awards OCI-0725070 and ACI-1238993).

## Additional information

### Funding

| Funder | Grant reference number | Author |
|---|---|---|
| National Institute of General Medical Sciences | R01GM085043 | Peijun Zhang |
| National Science Foundation | PHY-1430124 | Klaus Schulten |
| National Institute of General Medical Sciences | P50GM082251-7518 | Peijun Zhang |
| National Institute of General Medical Sciences | 9P41GM104601 | Klaus Schulten |
| National Institute of General Medical Sciences | 5R01GM098243 | Klaus Schulten |
| National Science Foundation | OCI-0725070 | Klaus Schulten |
| National Science Foundation | ACI-1238993 | Klaus Schulten |

The funders had no role in study design, data collection and interpretation, or the decision to submit the work for publication.

### Author contributions

CKC, BAH, FJA, Acquisition of data, Analysis and interpretation of data, Drafting or revising the article; JM, GZ, JRP, Acquisition of data, Analysis and interpretation of data; KS, Conception and design, Analysis and interpretation of data, Drafting or revising the article, Contributed unpublished essential data or reagents; PZ, Conception and design, Acquisition of data, Analysis and interpretation of data, Drafting or revising the article

### Author ORCIDs

Peijun Zhang, http://orcid.org/0000-0003-1803-691X

## Additional files

### Supplementary files

• Supplementary file 1. Summary of protein-protein interactions at key interfaces of equilibrated *T. maritima* unit cell model. Residues participating in a given interface but not associated with particular partners are listed separately for each domain. Residues that interact significantly (>50% of frames) are listed as a pair in a separate row. Interactions unique to this study are listed in green. Where ambiguous, residue pairs involving a receptor bound to CheA-P5, CheW from a CheA-P4/CheW ring or CheW from a CheW-only ring are denoted with a (1), (2) or (3) respectively. ** Signifies interfaces taken directly from experimental structures. Recent references pertaining to each protein-protein interface are given.

### Major datasets

The following datasets were generated:

| Author(s) | Year | Dataset title | Dataset ID and/or URL | Database, license, and accessibility information |
|---|---|---|---|---|
| Cassidy CK, Himes BA, Alvarez FJ, Ma J, Zhao G, Perilla JR, Shulten K, Zhang P | 2015 | Structure of bacterial chemotaxis signaling CheA2-trimer core complex by cryo-electron tomography and subvolume averaging | http://www.ebi.ac.uk/pdbe/entry/emdb/EMD-6319 | Publicly available at the EBI Protein Data Bank (Accession no: EMD-6319). |
| Cassidy CK, Himes BA, Alvarez FJ, Ma J, Zhao G, Perilla JR, Schulten K, Zhang P | 2015 | Structure of bacterial chemotaxis signaling CheA2-hexamer core complex by cryo-electron tomography and subvolume averaging | http://www.ebi.ac.uk/pdbe/entry/emdb/EMD-6320 | Publicly available at the EBI Protein Data Bank (Accession no: EMD-6320). |
| Cassidy CK, Himes BA, Alvarez FJ, Ma J, Zhao G, Perilla JR, Schulten K, Zhang P | 2015 | Cryo-electron Tomography and All-atom Molecular Dynamics Simulations Reveal a Novel Kinase Conformational Switch in Bacterial Chemotaxis Signaling | http://www.rcsb.org/pdb/search/structidSearch.do?structureId=3JA6 | Publicly available at the RCSB Protein Data Bank (Accession no: PDB-3JA6). |
| Cassidy CK, Himes BA, Alvarez FJ, Ma J, Zhao G, Perilla JR, Schulten K, Zhang P | 2015 | Tomogram of the reconstituted monolayer of bacterial chemotaxis core signaling complex | http://emsearch.rutgers.edu/atlas/3234_summary.html | Publicly available at the Electron Microscopy Data Bank (Accession number EMD-3234). |

The following previously published datasets were used:

| Author(s) | Year | Dataset title | Dataset ID and/or URL | Database, license, and accessibility information |
|---|---|---|---|---|
| Park SY, Bilwes AM, Crane BR | 2006 | Crystal Structure of the Cytoplasmic Domain of a Bacterial Chemoreceptor from Thermotoga Maritima | http://www.rcsb.org/pdb/explore/explore.do?structureId=2ch7 | Publicly available at the RCSB Protein Data Bank (Accession no: 2CH7). |
| Bilwes AM, Alex LA, Crane BR, Simon MI | 1999 | Crystal Structure of CHEA-289, A Signal Transducing Histidine Kinase | http://www.rcsb.org/pdb/explore/explore.do?structureId=1b3q | Publicly available at the RCSB Protein Data Bank (Accession no: 1B3Q). |
| Li X, Bayas C, Bilwes AM, Crane BR | 2013 | The structure of a ternary complex between CheA domains P4 and P5 with CheW and with an unzipped fragment of TM14, a chemoreceptor analog from Thermotoga maritima | http://www.rcsb.org/pdb/explore/explore.do?structureId=4jpb | Publicly available at the RCSB Protein Data Bank (Accession no: 4JPB). |

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
