## [Decision Letter]

Thank you for submitting your work entitled "CryoEM and Computer Simulations Reveal
a Novel Kinase Conformational Switch in Bacterial Chemotaxis Signaling" for peer
review at *eLife*. Your submission has been evaluated by Michael Marletta
(Senior editor), a Reviewing editor, and three reviewers. There are several major issues
that need to be addressed before a final decision can be made.

The reviewers have discussed the reviews with one another and the Reviewing editor has
drafted this decision to help you prepare a revised submission.

Summary:

The manuscript describes the use of subtomogram averaging to determine the structure of
a reconstituted chemotactic signaling array at intermediate-resolution.

A complex has been assembled on lipid monolayers between (1) the cytoplasmic domain of
Tar chemotaxis receptor, (2) CheA, and (3) CheW. A cryo-tomography density map of the
resulting crystalline patches has been obtained, and sub-tomogram averaging has been
used to an estimated resolution of the core-signalling unit around 1.3 nm.
Flexible-fitting molecular dynamics has been used to dock known crystal structures of
these components into the density map. This model and the molecular dynamics simulations
suggest a new conformational switch.

Overall, this study defines the reconstituted arrays at an unprecedented resolution and
thereby confirms key features of the assembly that have been proposed in the past, but
not previously visualized – such as the CheW rings and the P3 and P4 domain positioning.
These are important contributions to the understanding of the array structure. However,
there are major concerns about the anisotropic character of the EM data and the possible
impact on the accuracy of the model. Moreover, the reliability and uniqueness of the fit
of the model to the EM data must be assessed in order to judge the quality of the
model.

The all-atom MD simulations, which are impressive in their size and duration, suggest a
provocative dynamical model of kinase regulation, although the accompanying biochemical
data are not highly supportive of the "dipping" mechanism, and further control
experiments should be performed. Moreover, it is not entirely clear how these
conformational changes are to be interpreted in terms of CheA activation and whether
there are other relevant changes.

Reviewer #1:

Essential revisions:

1) The elegant combination of an in vitro reconstitution together with sub-tomogram
averaging and molecular modeling is very impressive, but this reviewer has two
substantial technical concerns: the data has been obtained by performing tomography on a
monolayer sample. All of the unit-cells are equivalently oriented relative to the
electron beam and during tomography are tilted through a limited angular range. As is
also the case in 2D electron-crystallography experiments, this results in a missing cone
of information in the final structure. The resolution of the structure is therefore
lower in Z than in X, Y. This situation cannot be avoided for a monolayer sample. This
anisotropic resolution has multiple implications: it will lead to apparent smearing of
the structure in Z; it will influence the ability to place the subunits into the map,
and especially to determine their Z-position; it will affect the MDFF. The authors only
quote a single resolution suggesting this effect may not be considered. What is the
resolution in Z? Does this allow reliable positioning of the structural models in Z? How
reliable is this positioning? How was the anisotropic resolution taken into account in
MDFF?

2) Knowing how the available structures have been placed into the EM map, and with what
confidence the positions are determined, is essential for assessing the reliability of
the final model. It is not clear to me how reliable these positions are. For example,
the authors state that the four-helix bundle of the CheA-P3 bundle is clearly discerned
to run parallel to the receptor. How clearly? Judging from the available supplementary
movie, there is some room for varying this position, as there is for the position of the
P5 domain, in particular in the Z direction. What is currently missing is an assessment
of how reliable the subunit models are and how accurately the EM map determines their
position and arrangement. Are any alternative arrangements possible? With what
confidence can this arrangement be selected? How accurately can the domain positions and
orientations be determined using an EM map of this resolution and these structural
subunit models? How accurate do these initial positions need to be to interpret details
after MDFF? One can imagine that with a 30 Å resolution structure, subsequent MDFF would
not give a structure reliable to side-chain resolution, while with a 5 Å structure it
would – what about at the resolution in this manuscript? In the end the authors go on to
determine specific amino acid interactions and conformational changes, so it is
important to clearly justify/demonstrate that this level of interpretation is reliable
and unique given the input data.

3) Example data should be deposited in the EMDB.

Reviewer #2:

Minor revisions:

1) Various CheA domains are first referred to in the subsection “3D density maps of
CheA_2_-timer and CheA_2_-hexamer”. It would be helpful if there
had been a sentence or two first to introduce the structure of CheA, identifying the
domains P1 to P5.

2) The authors rightly point out that the model agrees with previous structural studies,
and they go on to validate, in cells, the conformational change that they propose for
CheA. This should all be balanced by some discussion of what might be added by further
improving the resolution. While docking known structures into EM maps at a resolution of
~1.2 nm is generally accepted as giving useful biological information, it generally is
not accepted as giving a definitive final structure.

Reviewer #3:

Essential revisions:

1) The authors should provide more detail with respect to the uniqueness of the P4
positioning in the density maps. This is an important issue because the EM data here
alone determines the P4 position and it does not draw on crystal structures, such as
most of the other interfaces. The conformation of P4 is also critical to the dynamics
simulations that follow. The domain fitting was done reasonably, however; on viewing the
density, one wonders if other positions are also nearly as compatible. For example, do
the fits converge if different starting conformations are used? Could some metric be
given with respect to the goodness of fit? Also on this point, it appears that the
nucleotide-free form on the P4 domain was taken, both for the modeling and the dynamics.
ATP binding substantially affects the conformation of a large loop on the P4 domain
surface. Does ATP-bound P4 give a similar fit as the nucleotide-free form?

2) There is cryoEM data that suggests the P1 and P2 associate quite strongly with the
kinase core in the inhibited state (Briegel et al. 2013). If these regions indeed
contact P4, they could alter its positioning and dynamics. This point should be
noted.

3) This reviewer is concerned that the significance of the R297 salt bridge to
stabilizing the dipped conformation is overstated. In the simulations, is 4 versus 2
instances of dipping significant for 27 copies? Also, although 3 of the 4 dwell times
for the dipped conformation are indeed longer in the wt, the other one is similar to
that seen in the mutant. Is the principle component associated with this motion the only
one that shows a difference in occurrence between the wt and mutant trajectories? A more
thorough assessment of this data would bring more confidence to the dipping result.

4) The use of "asymmetry" might be misleading to differentiate the
conformational states. The term certainly describes the dipped state, however; the
asymmetry may simply be a consequence of this being a rare event. The reader gets the
impression that there are two states: one symmetric and one asymmetric; however, a
dipped symmetric state may also be possible, just unlikely to be seen in the same dimer
under the simulation conditions.

5) Little information is given with respect to the PCA in the dynamics trajectories (and
there is no mention of it in the Methods section). Also, for a general biological
scientist, the significance of Figure 5 will be
lost without a better description of what is being represented. It would be useful to
know what percent of the eigenvalues are represented by the first component and how many
vectors represent a majority of the total motion. Also, the authors focus on this one
dipping transition; are there no other significant conformational changes? The arrays
are of great interest in part due to their cooperative behavior; one might expect such
simulations to reveal recoupling across core particles. There is also data that changes
in the P5-receptor interfaces are associated with activation (Piasta et al. 2013). Are
any such motions observed? This should be commented on.

6) The cross-linking data supports proximity of the CheA receptor residues, but it does
not provide much support for the dipping motion. Control sites would have to be
investigated and relative rates of cross-linking determined, ideally for multiple
positions. For example, the R394 self cross-link seems to form as readily as
CheA-receptor cross-links. R394 self cross-links are not surprising given the symmetry
in the trimers-of-dimers, nonetheless, these sites are ~15 Å apart. Furthermore, it is
not clear why the N405 self cross-links also do not form, given that these residues are
closer to each other than are copies of R394. The statement near the end of the second
paragraph of the subsection “Biochemical validation of CheA conformational change in
*E. coli* cells” is not accurate, given that the yields of the 394
self cross-links are quite different in the two experiments. Normalizing to this band,
there is not much change between the 394/361 and 394/316 pairs. Even if the
cross-linking is less for 316/394, it could mean that those resides are simply further
apart in a dominant conformation or show reactivity differences (e.g. 405 vs. 394).
There is also the question of what state the cells are in during cross-linking. The
attractant serine is present during the experiment, but depending on the timing of the
washes, the cells are likely adapted. How much of CheA is activated and how much is
inhibited in these experiments? For only a few sites, as investigated here, it would be
more powerful to detect an attractant depend effect on relative cross-linking of the
reporters. As it stands, the cross-linking data primarily supports the general
architecture, but does not make the dipping mechanism more compelling.

7) The cell swimming assays need more description and from Figure 6 alone, it is not clear what parameter is being graphed.
"Swimming ability" itself is a misnomer, as the cells can likely swim fine,
but can't alternate tumbling properly to migrate in soft agar. This should be explained
in the text or figure legend. Do all the mutants form attractant rings at the migration
fronts indicative of chemotaxis? Or do some just show spreading, a consequence of some
CheA activation, but no regulation (pseudotaxis). Photographs of the swarm plates for
key mutants (or those that show differences should be provided as figure supplements).
Furthermore, simply measuring the swarm radius does not distinguish a mechanistic defect
from an assembly defect. Can an assessment be made (perhaps using the reconstitution
system) if the mutants form arrays like wt?

8) In the subsection “3D density maps of CheA_2_-timer and
CheA_2_-hexamer“, second paragraph, the electron density is less in the CheW/P5
contacts between neighboring core signaling particles than within the core complex, but
this does not necessarily mean that the interfaces are weaker; they may instead be more
structurally variable, and, hence, averaged out. Consider a change of wording. It is
also difficult to see in Figure 3 that
there is less density in the P5/W contacts between core particles. In Figure 3, color P5 and W differently, and demark the
interfaces 1 and 2 that are being referred to in the text.

9) This reviewer appreciates that the authors are being cautious with the interpretation
of their data, but the reader is largely left to summarize what the findings mean for
CheA activation. The mechanistic take home message should be more explicit.

[Editors' note: further revisions were requested prior to acceptance, as described
below.]

Thank you for submitting your work entitled "CryoEM and Computer Simulations Reveal
a Novel Kinase Conformational Switch in Bacterial Chemotaxis Signaling" for
consideration by *eLife*. Your article has been reviewed by three peer
reviewers, and the evaluation has been overseen by a Reviewing Editor and Michael
Marletta as the Senior Editor.

The reviewers have discussed the reviews with one another and the Reviewing editor has
drafted this decision to help you prepare a revised submission. The following comments
need to be addressed before a final decision can be made.

Reviewer #1:

1) Anisotropic resolution of the map.

The authors present the angular distribution of the contributing subunits and argue that
because is a good coverage of the tomography tilt range this concern can be disregarded.
This is not a valid argument for two reasons. Firstly because the angular coverage is
not good – high tilt angles are clearly underrepresented relative to low tilt angles,
and this will cause a resolution anisotropy (it is not completeness of coverage, but
uniformity of coverage that determines resolution anisotropy). Secondly, at high tilt
the monolayer sample is much thicker and the projections have lower signal to noise.
High and low tilts do not contribute equivalently to the final structure. This will also
cause resolution anisotropy. I reiterate my initial concern – that the resolution of the
structure is expected to be lower in Z than in X and Y. In the response to reviewers the
authors show some isosurfaces with simulated wedges. It is not possible to sensibly
assess the extent of anisotropic resolution from such images, and anyway, fitting is not
based on isosurfaces, but on density. I am not persuaded that resolution anisotropy can
be largely disregarded as the authors suggest.

This is not an unusual problem and it should not be difficult for the authors to address
this issue better. They should be clear in the manuscript that there is resolution
anisotropy and make an attempt to measure its extent (previous publications have used
for example FSC within cones or 3DSSR to assess anisotropic resolution). The potential
effect of this on fitting and MDFF should then be considered. Ideally, this would be
taken into account during simulation, but the authors will probably consider this too
much work. In that case they should write that this was not taken into account, and
discuss what the potential influence might be on the reliability of domain positioning
or on the simulations.

2) Reliability and uniqueness of fit.

This question was also asked by reviewer 3. How well does the EM density determine the
position of the domains, in particular P4? From the authors response to reviewers it
sounds like the position is not well defined, but that different positions still undergo
dipping motions. I am still concerned that the orientation of P4 is not well defined by
the EM map, that the starting orientation is critical to the MDFF analysis, and that
this influences the interpretation of the dipping motion. The authors have access to the
raw data and do not seem to be concerned by this. In that case they need to help the
reader to assess the reliability of the model. Ideally by providing a metric for the
goodness of fit, but minimally by an honest appraisal of the caveats, clearly and openly
discussing the reliability of the fit, where alternative positions are possible, and how
this might influence the interpretation. This should be in the main text of the
manuscript.

Reviewer #2:

The revisions made in response to my comments are fully satisfactory.

Reviewer #3:

In their revised version of the manuscript, the authors have done a nice job of
improving the paper and they have largely addressed my concerns. The one exception is
the cross-linking studies. I still do not agree with the statement that "Taken
together, our cross-linking experiments suggest that the CheA-P4 "dipped"
conformation observed in silico is indeed sampled within the native chemosensory complex
of *E. coli*." The inference here is that the cross-linking data
supports the dipped conformation as a relevant state of the arrays. It may well be, but,
the cross-linking data does not provide strong support for this supposition. Again, the
only band to report on the dipped conformation (361/394) is quite weak, much weaker than
the self 394-394 band. It's not uncommon to see some cross-linking between residues that
are in proximity. In fact, I would be surprised if no cross-linking was observed, even
if only the undipped conformation was represented. The authors can lean more on their
mutagenesis data to support the dipped conformation, but I don't think the cross-linking
makes the case.

---

## [Author Response]

[…] Overall, this study defines the reconstituted arrays at an unprecedented
resolution and thereby confirms key features of the assembly that have been proposed
in the past, but not previously visualized – such as the CheW rings and the P3 and P4
domain positioning. These are important contributions to the understanding of the
array structure. However, there are major concerns about the anisotropic character of
the EM data and the possible impact on the accuracy of the model.

Because our monolayer samples are not flat like 2D crystals, but are rather wavy, the
angular distribution of contributing subunits is very well sampled (Figure 3—figure supplement 1, see Reviewer #1, point 1),
thus anisotropic resolution of the density map is not a significant concern.
Nonetheless, we tested the effect of missing cone on the density map. As you can see in
Figure 6, there is little
density distortion, with a cone semi-angle of ~12°, a conservative estimate for the
sampling in this study (Figure 6,
green mesh), thus its impact on the accuracy of the model docking is minimal.

Author response image 1.****Effect of missing cone on the tomography density map.Shown in gray solid surface, the ideal density map without missing cone, and
maps with different amount of missing cone applied (colored meshes). Left,
green mesh 12° cone (this study, Figure 3—figure supplement 1). Right, yellow mesh, 30° cone as would be
found in other typical tomography studies. Density maps are all contoured at
1.5σ.**DOI:**
http://dx.doi.org/10.7554/eLife.08419.024

Moreover, the reliability and uniqueness of the fit of the model to the EM data
must be assessed in order to judge the quality of the model.

We would like to clarify that the unit cell model used in the bulk of our MD simulation
was built heuristically, utilizing existing X-ray structures to best preserve the known
interaction interfaces and only indirectly using the density maps to provide loose
constraints on portions of the model in which previous data did not exist, in particular
the position of the CheA P4 domains. This seems to have been a major source of confusion
and concern for the reviewers, and we apologize for not making it clearer. We have
revised the Methods section to include detailed steps on how the MD models were
generated with a new Figure 4—figure supplement 2. Nevertheless, we carried out additional equilibrium simulations of the
MDFF-refined CheA_2_-trimer model, which exhibit qualitatively the same
structural and dynamical features, including the presence of several CheA P4 dips, as
shown in the new Figure 4—figure supplement 5
(See Reviewer #1, Point 2 and Reviewer #3, Point 1). Moreover, to test the uniqueness of
our model fit, we carried out several additional fitting experiments for comparison,
obtaining virtually identical models (See Reviewer #3, Point 1 and Figure 7).

Author response image 2.****CheA dimers from multiple initial conformations converge to same
fitted model.Red, blue, and green traces track RMSD of CheA dimers as measured from the
initial MDFF-refined model. Though, the dimer conformations diversity during
the 120 ns production run, a 20 ns MDFF simulations returns them to their
initial MDFF-refined state.**DOI:**
http://dx.doi.org/10.7554/eLife.08419.025

The all-atom MD simulations, which are impressive in their size and duration,
suggest a provocative dynamical model of kinase regulation, although the accompanying
biochemical data are not highly supportive of the "dipping" mechanism, and
further control experiments should be performed.

We designed three cross-linking pairs for the predicted interactions based on the MD
model: the interaction only in the dipping state and those only in the undipped state.
The cross-linking results agree very well with model prediction, supporting our
computational model as well as confirming the existence of a novel dipping state (See
Reviewer #3, Point 6). We have included additional control experiments as presented in
Figure 5.

We also believe *this first plausible model with sufficient details of the basic
chemotaxis core-signaling unit* will be valuable for researchers in the field
to further test and investigate the function and mechanisms of the bacterial chemotaxis
signaling.

Moreover, it is not entirely clear how these conformational changes are to be
interpreted in terms of CheA activation and whether there are other relevant
changes.

We have added discussions on the P4 conformation in terms of CheA activation
(Discussion). The dipping motion observed in MD simulation warrants further in depth
investigation, such as how these conformational changes are related to CheA activation.
This is beyond the scope of this study.

Detailed responses to all points raised by the individual reviewers are provided
below.

Reviewer #1:

Essential revisions:1) The elegant combination of an in vitro reconstitution
together with sub-tomogram averaging and molecular modeling is very impressive, but
this reviewer has two substantial technical concerns: the data has been obtained by
performing tomography on a monolayer sample. All of the unit-cells are equivalently
oriented relative to the electron beam and during tomography are tilted through a
limited angular range. As is also the case in 2D electron-crystallography
experiments, this results in a missing cone of information in the final structure.
The resolution of the structure is therefore lower in Z than in X, Y. This situation
cannot be avoided for a monolayer sample. This anisotropic resolution has multiple
implications: it will lead to apparent smearing of the structure in Z; it will
influence the ability to place the subunits into the map, and especially to determine
their Z-position; it will affect the MDFF. The authors only quote a single resolution
suggesting this effect may not be considered. What is the resolution in Z? Does this
allow reliable positioning of the structural models in Z? How reliable is this
positioning? How was the anisotropic resolution taken into account in
MDFF?

We thank the reviewer for the positive remark, and agree with the reviewer that a
perfectly oriented sample, such as 2D crystal, has a substantial anisotropic resolution
(poorer in Z) with a missing cone of information in the final structure. However, our
monolayer samples are not rigid like 2D crystals, but are quite wavy on the cryoEM
grids. The angles between the normal of the monolayer and the electron beam can be as
much as 15°. On top of this, there is additional specimen pre-tilt ranging 0-10°, in
addition to a tomography tilt range ± 70°. The overall angular distribution of all the
subunits contributing to the averaged map is presented in a new Figure 3—figure supplement 1 and C for the in-plane rotation and
maximum tilt angle, respectively. The plot shows a near uniform distribution of in-plane
angles, and a very good coverage of the tomography tilt range, therefore, concerns with
missing cone and anisotropic resolution, as well as its impact on the reliability of
positioning of MDFF model can be largely disregarded (see overall critique point #1,
Figure 6).

2) Knowing how the available structures have been placed into the EM map, and
with what confidence the positions are determined, is essential for assessing the
reliability of the final model. It is not clear to me how reliable these positions
are. For example, the authors state that the four-helix bundle of the CheA-P3 bundle
is clearly discerned to run parallel to the receptor. How clearly? Judging from the
available supplementary movie, there is some room for varying this position, as there
is for the position of the P5 domain, in particular in the Z direction. What is
currently missing is an assessment of how reliable the subunit models are and how
accurately the EM map determines their position and arrangement. Are any alternative
arrangements possible? With what confidence can this arrangement be selected? How
accurately can the domain positions and orientations be determined using an EM map of
this resolution and these structural subunit models? How accurate do these initial
positions need to be to interpret details after MDFF? One can imagine that with a 30
Å resolution structure, subsequent MDFF would not give a structure reliable to
side-chain resolution, while with a 5 Å structure it would – what about at the
resolution in this manuscript? In the end the authors go on to determine specific
amino acid interactions and conformational changes, so it is important to clearly
justify/demonstrate that this level of interpretation is reliable and unique given
the input data.

We understand the reviewer’s concerns and have taken a number of steps to better
characterize to what degree our models are unique and robust. These steps are discussed
in more detail below (also see Reviewer #3, Point 1) and the modeling procedures used in
this study, including “how” the available structures were placed in the EM maps, are
summarized in a new supplemental figure (Figure 4—figure supplement 3). Here, we will address some of the reviewer’s more
general concerns.

Regarding how reliably the EM map determines the positions and arrangement of the
protein components involved in the model, we can only strictly state that our MDFF model
is reliable up to a resolution equal to that of the lowest resolution piece used in its
construction, that is our 12.7 Å density map. Nevertheless, this resolution places tight
constraints on the overall positioning and tertiary structure of the proteins involved.
In addition, we would like to highlight that the proteins were not docked into the EM
map in one-by-one but rather heuristically-constructed, full subunit pieces were docked
to best preserve known interfaces (e.g., receptors within receptor TOD from 1QU7,
CheA-P5/CheW interfaces from 4JPB, receptor/CheW and receptor/CheA-P5 from NMR, etc.).
Assuming the correctness of these previously-characterized interfaces, no other global
arrangements are possible. To illustrate this we conducted additional MDFF simulations
on the MDFF-refined CheA_2_-trimer model increasing the coupling of the model
to the map by an order of magnitude – a regime in which the density-derived forces
overwhelm the electrostatic forces between atoms.

Concerning the interpretation of structural details after MDFF, the reviewer is indeed
correct that at the resolutions of the maps present in this study, no side-chain
information is present and, hence, MDFF cannot directly bias the structures at the
side-chain level. However, we would like to highlight that the strength of the MDFF
methodology is the “flexibility” of its fitting. This flexibility arises from the use of
the chemically-based MD force field, which adds much additional information to help
refine the model’s side chain orientations and interactions during the fitting
procedure. Unfortunately, there currently are no robust methods for assessing the
“accuracy” (that is, how closely the model matches the “true” structure) of a model
derived via the fitting of high-resolution structures into lower-resolution EM maps,
especially those with resolution lower than 8 Å. Indeed, this is an area of active
research in the field (Xu, X. P., & Volkmann, N. (2015) Arch. Biochem. Biophys.;
Schröder, G. F. (2015) C.O.S.B.). Rather, validation of the reliability of a model may
still be given in terms of its “precision”, that is the qualitative reproducibility of
the model’s key predictions. Importantly, we subsequently carried out extensive,
*all-atom* MD simulations on multiple models and using many
independent simulations and demonstrated that the key discovery of this study, namely
the dipping conformational dynamics of the CheA kinase domain, is reproduced in multiple
simulations of both heuristically- and MDFF-derived models. We then successfully
confirmed a number of our model’s predictions in the biochemical experiments, the best
test of reliability.

Finally, the authors would also like to note that while the CheA P3 and P5 domains may
appear in the supplementary movie to have more room for varying their positions, this
appearance is a consequence of the chosen isosurface used to illustrate a rough envelope
of the electron density. This envelope can appear tighter or more diffuse depending on
the isosurface value chosen. The MDFF biasing force, on the other hand, is based on the
actual three-dimensional volumetric density, not the envelope.

3) Example data should be deposited in the EMDB.

The EM maps were deposited in EMDB under accession code EMD-6319 for the
CheA_2_-trimer and EMD-6320 for the CheA_2_-hexamer, at the time
when the manuscript was submitted. The EM map-associated MD model of the *T.
maritima* core-signaling unit was also deposited in PDB database under
accession code 3JA6.

Reviewer #2:

Minor revisions:

1) Various CheA domains are first referred to in the subsection “3D density maps
of CheA_2_-timer and CheA_2_-hexamer”. It would be helpful if there
had been a sentence or two first to introduce the structure of CheA, identifying the
domains P1 to P5.

We thank the reviewer’s suggestion and have included a sentence to introduce the
structure of CheA, identifying the domains P1 to P5 (“CheA is a multi-domain protein,
consisting of five separate and functionally distinct domains (P1-P5): P1-phosphoryl
transfer domain, P2-substrate binding domain, P3-dimerization domain, P4-kinase domain
and P5-regulatory domain”).

2) The authors rightly point out that the model agrees with previous structural
studies, and they go on to validate, in cells, the conformational change that they
propose for CheA. This should all be balanced by some discussion of what might be
added by further improving the resolution. While docking known structures into EM
maps at a resolution of ~1.2 nm is generally accepted as giving useful biological
information, it generally is not accepted as giving a definitive final
structure.

We appreciate the reviewer’s comments and have added a discussion regarding what one
might expect from modeling efforts utilizing higher resolution EM maps (“By further
improving the resolution of our cryoET to better than 8Å and by using the novel
lipid-monolayer system described here, an atomic homology model for *E.
coli* can be derived, which would greatly facilitate the use of the wealth of
existing biochemical and biophysical data and permit directly transferrable structural
and dynamical predictions”).

Reviewer #3:

Essential revisions:1) The authors should provide more detail with respect to
the uniqueness of the P4 positioning in the density maps. This is an important issue
because the EM data here alone determines the P4 position and it does not draw on
crystal structures, such as most of the other interfaces. The conformation of P4 is
also critical to the dynamics simulations that follow. The domain fitting was done
reasonably, however; on viewing the density, one wonders if other positions are also
nearly as compatible. For example, do the fits converge if different starting
conformations are used? Could some metric be given with respect to the goodness of
fit? Also on this point, it appears that the nucleotide-free form on the P4 domain
was taken, both for the modeling and the dynamics. ATP binding substantially affects
the conformation of a large loop on the P4 domain surface. Does ATP-bound P4 give a
similar fit as the nucleotide-free form?

We thank the reviewer for the well-thought-out questions regarding the overall quality
and uniqueness of our models. We have added additional text and figures, as described
below, to address the reviewer’s concerns.

To address the uniqueness of the positioning of the CheA P4 domain and its effect on the
observed P4 dynamics, we have carried out an additional series of extensive simulations
of the MDFF-refined CheA_2_-trimer model (Figure 4—figure supplement 3 and [Supplementary-material SD1-data]) for comparison with the dynamics observed
in the heuristically-derived unit cell model. As the MDFF refinement procedure affected,
in particular, the positioning of the P4 domains relative to the rest of the CheA dimer
molecules and receptor tips, and hence their interactions with neighboring proteins, the
two models provided different CheA dimer initial conditions from which to seed further
simulations. In particular, the MDFF fitting procedure gave rise to a CheA dimer model
with an RMSD from the heuristically modeled CheA dimer of ~5.3 Å. Nevertheless, both
models exhibited multiple dipping events (Figure 4 and Figure 4—figure supplement 5)
demonstrating that the ability of the P4 domain to undergo the dipping motion does not
depend on a more precise placement of the P4 domains than our density maps have
provided.

In addition, to address the convergence of the MDFF fit, we conducted two additional
MDFF simulation on CheA_2_-trimer model in which the CheA molecules exhibited a
mix of P4 conformations. For this purpose we selected as initial conformations two of
the ten CheA_2_-trimer replicas (subsequent to the ~120 ns production
simulation) in which one or more of the CheA molecules exhibited dipped or
“intermediate” P4 conformations. Figure 7 depicts, for one of those MDFF simulations, the RMSD of each CheA dimer in
the CheA_2_-trimer (as measured from the initial MDFF-refined CheA dimer model)
over the course of the ~120 ns production simulations and 20 ns post-production
simulation. Hence, we show that the MDFF fitting of CheA dimers starting from diverse
initial configurations all converge on to essentially the same model.

Finally, the reviewer is correct that the nucleotide-free form of CheA was selected for
this initial study. In general, maps of resolution less than 1 nm cannot be used to
discern differences in secondary structure (Schröder, G. F. (2015). COSB, 31, 20-27).
Regarding the structural effects of nucleotide binding on the local structure of the P4
domain because both conformations inhabit essentially the same volume, we cannot, at the
current resolution of our maps (>=12.7Å), discriminate with MDFF the difference in
conformation of the loop in the nucleotide-free P4 (e.g. PDB 1B3Q) and nucleotide-bound
P4 (e.g., PDB 1I58 or 1I5D).

2) There is cryoEM data that suggests the P1 and P2 associate quite strongly
with the kinase core in the inhibited state (Briegel et al. 2013). If these regions
indeed contact P4, they could alter its positioning and dynamics. This point should
be noted.

This was indeed carefully considered when we docked the models. We now have a
preliminary density map lacking the P1-P2 domains in the CheA. The map with CheA P345 is
very similar to the map with CheA full length (Figure 8), further suggesting the density corresponds to the P4
domain. The P1-P2 domains are likely very flexible and the densities have been averaged
out. The main differences between this study and the previous one (Briegel et al. 2013)
are 1) the map resolution is better in this study and number of subunits contributing to
the final structure is in a great excess; 2) our density map is from wt receptor, likely
in the adapted state, while the other study is in the inhibited state.

Author response image 3.****Comparison of density maps containing full-length wt CheA and wt
CheA-P345.Densities are contoured at 2σ. Solid blue, with wt CheA-P345; red mesh, with
full-length wt CheA. The density with CheA-P345 appears elongated along the Z
direction, due to the missing wedge effect and limited amount of data.
(**A**) Side view of the volume slab. (B and C) Top views of volume
slabs at the height indicated in A.**DOI:**
http://dx.doi.org/10.7554/eLife.08419.026

3) This reviewer is concerned that the significance of the R297 salt bridge to
stabilizing the dipped conformation is overstated. In the simulations, is 4 versus 2
instances of dipping significant for 27 copies? Also, although 3 of the 4 dwell times
for the dipped conformation are indeed longer in the wt, the other one is similar to
that seen in the mutant. Is the principle component associated with this motion the
only one that shows a difference in occurrence between the wt and mutant
trajectories? A more thorough assessment of this data would bring more confidence to
the dipping result.

We thank the reviewer for this careful observation. After performing additional
experiments described below, we agree that while the R297/E397 interaction clearly
stabilizes the “dipped” state in silico as signified by an increased dwell time and is
also critical for proper signaling in vivo, this does not necessarily mean that it is
the stabilization of the “dipped” state that makes the interaction critical for proper
signaling. For example, one could imagine the interaction being related to a Class I
model given the spatial relationship with this pair and R354/E392 examined in a recent
NMR study. On the other hand, it could also be that the rearrangement of P4 is important
to bring into register binding sites for P1 in support of a class II model, or some
combination of the two. We have altered the main text to clarify that the R297/E397
contact is necessary for signaling and correlated to the “dipped” state, but softened
the correlation between the two ideas.

To address this concern experimentally, we extended both our wild-type and R297A mutant
unit cell simulation by 50%, from 300 ns each to 450 ns each, in order to improve our
sampling. We did not observe any additional stabilized dipping events in the wild-type
simulations (Figure 4, top). Moreover, in the
R297A mutant simulations, one of the dipping CheA P4s underwent an additional dip-return
event (Figure 4, bottom), further closing the
gap between the sheer number of dipping events observed in the two models. Hence, in
light of the new data, we agree with the reviewer that our proposal that the R297/E397
contact may help promote the dipping conformation change is not well supported. We have
removed this notion from the main text. On the other hand, our suggestion that the
R297/E397 contact may help stabilize the dipping conformation seems to be reinforced by
our extended simulations. In particular, in the wild-type simulations, the three (of the
four total) dipping CheA dimers in which the R297/E397 contact was formed (Figure 4, top, red, blue, green traces) maintain
this conformation, while the dimer that did not form the R297/E397 contact (yellow
trace) returned to the bulk. In addition, both dipping CheA dimers in the R297A mutant
simulations, which are of course unable to from the R297/E397 salt bridge, quickly
return to the bulk (Figure 4, bottom, red, blue
traces), including the P4 domain, which undergoes two dip-return events (blue trace).
Hence, we feel the suggestion of the stabilizing nature of the R297/E397 contact is
reasonable in light of the comparative in silicodata. It remains to be determined if the
stabilization of the “dipped” state is the primary role of R297 in signaling, or if this
is an ancillary effect.

Finally, regarding the differences between the PC projections of the wt and R297A mutant
CheA dimer populations, the authors wish to point out that the PCA was constructed to
highlight the dip for the purpose of tracking the CheA dimer conformations. Hence, the
difference between the CheA dimer dynamics of the two models is most readily observed in
the first principal component, which captures a large percent (41%) of this dipping
motion. Though the second PC also describes a large portion of the total variance (31%),
the two models do not differ significantly from each other in this dimension.

4) The use of "asymmetry" might be misleading to differentiate the
conformational states. The term certainly describes the dipped state, however; the
asymmetry may simply be a consequence of this being a rare event. The reader gets the
impression that there are two states: one symmetric and one asymmetric; however, a
dipped symmetric state may also be possible, just unlikely to be seen in the same
dimer under the simulation conditions.

We agree with the reviewer and have changed the term "asymmetry" to
"dipped".

5) Little information is given with respect to the PCA in the dynamics
trajectories (and there is no mention of it in the Methods section). Also, for a
general biological scientist, the significance of Figure 5 will be lost without a better description of what is being
represented. It would be useful to know what percent of the eigenvalues are
represented by the first component and how many vectors represent a majority of the
total motion. Also, the authors focus on this one dipping transition; are there no
other significant conformational changes? The arrays are of great interest in part
due to their cooperative behavior; one might expect such simulations to reveal
recoupling across core particles. There is also data that changes in the P5-receptor
interfaces are associated with activation (Piasta et al. 2013). Are any such motions
observed? This should be commented on.

We appreciate the reviewer’s comments and have added to the Methods section an entry
entitled “Simulation Analysis” to describe the PCA procedure in more detail. In that
section the fractional variances of the first three modes are given, in particular, the
first component is shown to describe roughly 42% of the total motion. In addition, a
more detailed main-text discussion of the PCA results was added to better describe their
significance (subsection “A conformational change of the CheA kinase domain”, last
paragraph). Regarding the observation of “other” significant conformation changes, we
agree with the reviewer that our model and methods should be able to provide useful
information regarding general motions of the array, including those potentially relevant
to its cooperative nature. Indeed, we have sought to develop the first plausible
all-atom model for asking these kinds of structural questions and have deposited this
model in the PDB database for review and use by other researchers in the field. In this
study, however, we have chosen to focus our investigation on a large-scale dipping
motion of the CheA-P4 domain. Additional investigation will be necessary to test the
full extent of our model’s predictability in other areas of signaling and function, in
particular the P5-receptor interface mentioned by the reviewer is of active interest to
us.

*6) The cross-linking data supports proximity of the CheA receptor residues, but
it does not provide much support for the dipping motion. Control sites would have to
be investigated and relative rates of cross-linking determined, ideally for multiple
positions. For example, the R394 self cross-link seems to form as readily as
CheA-receptor cross-links. R394 self cross-links are not surprising given the
symmetry in the trimers-of-dimers, nonetheless, these sites are ~15 Å apart.
Furthermore, it is not clear why the N405 self cross-links also do not form, given
that these residues are closer to each other than are copies of R394. The statement
near the end of the second paragraph of the subsection “Biochemical validation of
CheA conformational change in* E. coli *cells” is not accurate, given
that the yields of the 394 self cross-links are quite different in the two
experiments. Normalizing to this band, there is not much change between the 394/361
and 394/316 pairs. Even if the cross-linking is less for 316/394, it could mean that
those resides are simply further apart in a dominant conformation or show reactivity
differences (e.g. 405 vs. 394). There is also the question of what state the cells
are in during cross-linking. The attractant serine is present during the experiment,
but depending on the timing of the washes, the cells are likely adapted. How much of
CheA is activated and how much is inhibited in these experiments? For only a few
sites, as investigated here, it would be more powerful to detect an attractant depend
effect on relative cross-linking of the reporters. As it stands, the cross-linking
data primarily supports the general architecture, but does not make the dipping
mechanism more compelling.*

Although cross-linking experiments cannot catch the dynamic motion of dipping, the data
support the appearance of a new interaction interface upon P4 dipping (by close
proximity of the CheA – receptor residues), that otherwise would not be formed. Although
limited in number, we tested the sites for three scenarios: exists in both states as
positive control (I304/N405); only in the undipped state (D316/R394); and only in dipped
state (E361/R394), along with necessary negative controls. Regarding the different
cross-linking yields (R394) in two different experiments, we have repeated all the
cross-linking in a single experimental setting, and updated the Figure 5 (previously Figure 6) with new data. Additional controls were also added in Figure 5.

We agree with the reviewer that low cross-linking intensity could mean that the residues
are farther apart in a dominant conformation or show reactivity differences, in addition
to low frequency of the conformation occurs. We have changed the discussion accordingly
to reflect this (subsection “Biochemical validation of CheA conformational change in
*E. coli* cells”, second paragraph).

Regarding the question about the state of the cells during cross-linking, the
experiments were carried out in the presence of serine, and the cells are most likely
adapted. What we measure is the ensemble of the bacterial population likely containing
both activated and inhibited CheA. We don’t know how much CheA is in each state. We
would like to point out that one of the major goals of this study is to put out*a
first plausible model with sufficient details of the basic chemotaxis core-signaling
unit* for researchers in the field. The cross-linking data support our
computational model and the probable novel dipping state. It would be beyond the scope
of this study to further detect signaling events with relative cross-linking of the
reporters, and define the mechanisms of the dipping conformation. We will certainly
follow up the leads and continue our investigation in this respect. Nevertheless,
following the reviewer’s suggestions, we carried out the experiments to test the effect
of serine on the existing cross-linking pairs under two different conditions: 1) cells
were grown in the presence of serine, followed by cross-linking with or without serine
washout; 2) cells were grown in the absence or presence of serine, followed by
cross-linking. In both cases, the changes in the extent of cross-linking are not
substantial or conclusive. A more thorough future investigation is needed.

7) The cell swimming assays need more description and from Figure 6 alone, it is not clear what parameter is being graphed.
"Swimming ability" itself is a misnomer, as the cells can likely swim fine,
but can't alternate tumbling properly to migrate in soft agar. This should be
explained in the text or figure legend. Do all the mutants form attractant rings at
the migration fronts indicative of chemotaxis? Or do some just show spreading, a
consequence of some CheA activation, but no regulation (pseudotaxis). Photographs of
the swarm plates for key mutants (or those that show differences should be provided
as figure supplements). Furthermore, simply measuring the swarm radius does not
distinguish a mechanistic defect from an assembly defect. Can an assessment be made
(perhaps using the reconstitution system) if the mutants form arrays like
wt?

We appreciate the reviewer’s comments. We have added detailed description of cell
swimming assays in the Methods section (subsection “Soft agar Assays”). We also included
the photographs of the representative soft agar plates for key mutants with controls in
the revised Figure 5. Furthermore, we carried
cryoEM analysis of the *E. coli* cells carrying these mutants and show
that these cells are functional in forming chemotaxis clusters. These results are
included in the new Figure 5—figure supplement 1.

8) In the subsection “3D density maps of CheA_2_-timer and
CheA_2_-hexamer“, second paragraph, the electron density is less in the
CheW/P5 contacts between neighboring core signaling particles than within the core
complex, but this does not necessarily mean that the interfaces are weaker; they may
instead be more structurally variable, and, hence, averaged out. Consider a change of
wording. It is also difficult to see in Figure 3 that there is less density in the P5/W contacts between core particles.
In Figure 3, color P5 and W differently, and
demark the interfaces 1 and 2 that are being referred to in the text.

We agree with the reviewer’s point and changed the wording in the revised manuscript
accordingly. We have clearly marked the interfaces 1 and 2 with arrows accordingly in
Figure 3, and used a darker surface
color to display these interfaces.

9) This reviewer appreciates that the authors are being cautious with the
interpretation of their data, but the reader is largely left to summarize what the
findings mean for CheA activation. The mechanistic take home message should be more
explicit.

We thank reviewer’s suggestion. We have added additional discussion on the implication
of our findings (Discussion).

[Editors' note: further revisions were requested prior to acceptance, as described
below.]

Reviewer #1:

1) Anisotropic resolution of the map.The authors present the angular
distribution of the contributing subunits and argue that because is a good coverage
of the tomography tilt range this concern can be disregarded. This is not a valid
argument for two reasons. Firstly because the angular coverage is not good – high
tilt angles are clearly underrepresented relative to low tilt angles, and this will
cause a resolution anisotropy (it is not completeness of coverage, but uniformity of
coverage that determines resolution anisotropy). Secondly, at high tilt the monolayer
sample is much thicker and the projections have lower signal to noise. High and low
tilts do not contribute equivalently to the final structure. This will also cause
resolution anisotropy. I reiterate my initial concern – that the resolution of the
structure is expected to be lower in Z than in X and Y. In the response to reviewers
the authors show some isosurfaces with simulated wedges. It is not possible to
sensibly assess the extent of anisotropic resolution from such images, and anyway,
fitting is not based on isosurfaces, but on density. I am not persuaded that
resolution anisotropy can be largely disregarded as the authors suggest.This is not
an unusual problem and it should not be difficult for the authors to address this
issue better. They should be clear in the manuscript that there is resolution
anisotropy and make an attempt to measure its extent (previous publications have used
for example FSC within cones or 3DSSR to assess anisotropic resolution). The
potential effect of this on fitting and MDFF should then be considered. Ideally, this
would be taken into account during simulation, but the authors will probably consider
this too much work. In that case they should write that this was not taken into
account, and discuss what the potential influence might be on the reliability of
domain positioning or on the simulations.

We agree with the reviewer that the resolution of the structure is poorer in Z than in X
and Y. To assess the degree of anisotropy in resolution, we calculated conical Fourier
shell correlations as suggested by the reviewer (see Figure 9). The overall resolution (FSC=0.143) is 11.3 Å
(calculated from the same map with an updated mask to include just the
CheA_2_-trimers, while the previous FSC was calculated using an extended mask),
and the resolution in Z direction is 15.8 Å. We have revised the manuscript to include
this in Figure 3—figure supplement 1. We
further repeated the MDFF exercise, taking anisotropic resolution into account during
simulation by using the anisotropic FSC low-pass filtered map. The resulting MDFF model
((Figure 9, orange) is similar
to the previous one (Figure 9,
yellow), with an overall RMSD of 2.78Å. The map and MDFF model deposited at PDB and EMDB
are being updated.

Author response image 4.****Resolution anisotropy.**(A)** Gold-standard Fourier shell correlation (FSC) of the
CheA_2_-trimer density map. The overall FSC of the map is plotted
as a solid black line. The FSC curves for the conical Fourier shells along the
X, Y, and Z directions are in solid green, blue and dark-red, respectively, and
along the 10 other directions are in dotted lines. The previous FSC (dashed
black line) was calculated from the same map but with a more extended mask.
**(B)** MDFF model of the CheA-P4 domain from the anisotropic FSC
filtered density map (orange), overlapped with the previous MDFF model
(yellow). The pink and light blue spheres indicate the P3-C and P5-N termini,
respectively. CheA-P3 and CheA-P5 domains are in gray.**DOI:**
http://dx.doi.org/10.7554/eLife.08419.027

2) Reliability and uniqueness of fit.This question was also asked by reviewer 3.
How well does the EM density determine the position of the domains, in particular P4?
From the authors response to reviewers it sounds like the position is not well
defined, but that different positions still undergo dipping motions. I am still
concerned that the orientation of P4 is not well defined by the EM map, that the
starting orientation is critical to the MDFF analysis, and that this influences the
interpretation of the dipping motion. The authors have access to the raw data and do
not seem to be concerned by this. In that case they need to help the reader to assess
the reliability of the model. Ideally by providing a metric for the goodness of fit,
but minimally by an honest appraisal of the caveats, clearly and openly discussing
the reliability of the fit, where alternative positions are possible, and how this
might influence the interpretation. This should be in the main text of the
manuscript.

We appreciate the reviewer’s comment and carried out additional docking exercises. We
fit the CheA-P4 domain via rigid-body transformations starting from 10,000 random
angular orientations and up to 20Å shifts from the center of the mass, to generate a
metric for the goodness of fit. This fitting resulted in 23 distinct classes (separated
by 3° and 3Å), as shown in Figure 10. The class of best fit (Figure 10, panel 1, with highest cross correlation and highest number of
contributing fit) is similar to the MDFF model (Figure 10, Panel 10 with overlay). In addition to the best fit, one
other possible fit is #5 (Figure 10, panel 5), given the proximity of the N and C termini of P4 to the P3 and
P5 domain. This fit is less probable compared to the best fit, considering that it has a
lower cross-correlation value, a much lower occupancy with ~ 1/3 the number of
contributing fits, and the positions of P4-N and C termini are reversed (flipped),
making it hard to connect the P3 and P5 termini with short linkers. In the revised
manuscript, we have included the fitting metric and a discussion on the possible
alternative P4 positions that might influence the interpretation (subsection “All-atom
model of the *T. maritima* chemosensory array”, Figure 3—figure supplement 3).

Author response image 5.A metric for the goodness of fit for the docking of the CheA-P4
domain.**(A)** Distribution of 23 classes of fits for the P4 domain starting
from random orientations. **(B)** The models from the top 9 highest
cross-correlation classes are shown in panels 1-9, with the cross-correlation
values and number of contributing fits (%) indicated below. These 9 classes
constitute 90.1% of total fits. Panel 10 is an overlay of the #1 fit (blue)
with the MDFF model (orange). The red and blue spheres indicate the N and C
termini of P4, respectively. The corresponding connecting termini from P3 and
P5 are in light blue and pink, respectively. P3 and P5 domains are in gray.**DOI:**
http://dx.doi.org/10.7554/eLife.08419.028

Reviewer #3:

*In their revised version of the manuscript, the authors have done a nice job of
improving the paper and they have largely addressed my concerns. The one exception is
the cross-linking studies. I still do not agree with the statement that "Taken
together, our cross-linking experiments suggest that the CheA-P4 "dipped"
conformation observed in silico is indeed sampled within the native chemosensory
complex of* E. coli*." The inference here is that the
cross-linking data supports the dipped conformation as a relevant state of the
arrays. It may well be, but, the cross-linking data does not provide strong support
for this supposition. Again, the only band to report on the dipped conformation
(361/394) is quite weak, much weaker than the self 394-394 band. It's not uncommon to
see some cross-linking between residues that are in proximity. In fact, I would be
surprised if no cross-linking was observed, even if only the undipped conformation
was represented. The authors can lean more on their mutagenesis data to support the
dipped conformation, but I don't think the cross-linking makes the case.*

We appreciate the reviewer’s comments. We have revised the manuscript to tone down the
inference from the cross-linking experiments and changed the statement to “suggesting
that the CheA-P4 "dipped" conformation observed in silico may have been
sampled within the native chemosensory complex of *E. coli*” (subsection
“Biochemical validation of CheA conformational change in *E. coli*
cells”, second paragraph).